# DynaTag for efficient mapping of transcription factors in low-input samples and at single-cell resolution

Pascal Hunold [1,2], Giulia Pizzolato[1,2], Nadia Heramvand[1,2], Laura Kaiser [2], Giulia Barbiera[3], Olivia van Ray[1,2], Roman Thomas [2,4], Julie George [2,5], Martin Peifer [1,2] & Robert Hänsel-Hertsch [1,2,6,7] ✉

Systematic discovery of transcription factor (TF) landscapes in low-input samples and at single cell level is a major challenge in the fields of molecular biology, genetics, and epigenetics. Here, we present cleavage under *Dyna*mic targets and *Tag*mentation (DynaTag), enabling robust mapping of TF-DNA interactions using a physiological salt solution during sample preparation. DynaTag uncovers occupancy alterations for 15 TFs in stem cell and cancer tissue models. We highlight changes in TF-DNA binding for NANOG, MYC, and OCT4, during stem-cell differentiation, at both bulk and single-cell resolutions. DynaTag surpasses CUT&RUN and ChIP-seq in signal-to-background ratio and resolution. Furthermore, using tumours of a small cell lung cancer model derived from a single female donor, DynaTag reveals increased chromatin occupancy of FOXA1, MYC, and the mutant p53 R248Q at enriched gene pathways (e.g. epithelial-mesenchymal transition), following chemotherapy treatment. Collectively, we believe that DynaTag represents a significant technological advancement, facilitating precise characterization of TF land-scapes across diverse biological systems and complex models.

Dynamic transcription factor (TF) interactions with DNA are crucial for gene regulation, cellular identity, development and diseases[1,2]. Aberrant TF activity is a prominent characteristic of various diseases, including cancer, where TFs often act as oncogenes or lose their native tumour-suppressive functions[3,4]. Mapping the genomic occupancy of TFs is therefore essential for elucidating gene regulatory networks, identifying molecular vulnerabilities and potential therapeutic targets. Chromatin immunoprecipitation followed by sequencing (ChIP-seq) has long been the gold standard for mapping protein–DNA interactions[5]. However, this method requires substantial input material for TFs, making it incompatible with single-

cell applications. Cleavage Under Targets and Release Using Nuclease (CUT&RUN) has emerged as an alternative approach, allowing for in situ profiling of chromatin interactions—including those involving TFs—with lower input requirements[6–8]. Despite this technical advancement, CUT&RUN requires extensive ligation-based library preparation and, to date, has not been applied to TFs at single-cell resolution[9]. Cleavage Under Targets and Tagmentation (CUT&Tag) employs protein A fused to the transposase Tn5 (pA-Tn5) as a probe to directly insert sequencing adaptors into DNA at the target sites. CUT&Tag enables robust mapping of histone–DNA interactions in situ within nuclei, in small samples, with single-cell

[1]Center for Molecular Medicine Cologne, Faculty of Medicine and University Hospital, Cologne, University of Cologne, Cologne, Germany. [2]Department of Translational Genomics, Faculty of Medicine and University Hospital Cologne, University of Cologne, Cologne, Germany. [3]Genevia Technologies Oy, Tampere, Finland. [4]University of Cologne, Faculty of Medicine and University Hospital Cologne, Institute of Pathology, Cologne, Germany. [5]Department of Otorhinolaryngology, Head and Neck Surgery, University Hospital of Cologne, Cologne, Germany. [6]Cologne Excellence Cluster on Cellular Stress Responses in Aging-Associated Diseases (CECAD), University of Cologne and University Hospital Cologne, Cologne, Germany. [7]Institute of Human Genetics, University Hospital Cologne, Cologne, Germany. ✉e-mail: robert.haensel-hertsch@uni-koeln.de

and spatial resolution[10,11]. However, employing CUT&Tag to map dynamic, low-affinity target–DNA interactions[12,13], such as those involving TFs, remains challenging, primarily due to difficulties in retaining these interactions during sample preparation. More recently, alternative tagmentation-based techniques, such as ACT-seq, CoBATCH[14], nano-CUT&Tag[15] and ChIL-seq[16] have emerged. Nevertheless, these technologies require the use of non-physiological, high-salt concentrations to suppress untargeted, false-positive tagmentation events. Importantly, TF–DNA interactions are sensitive to these stringent conditions, causing their dissociation from chromatin and incompatibility with these tagmentation-based technologies. To overcome these limitations, we present cleavage of Dynamic targets and Tagmentation (DynaTag), an adaptation of CUT&Tag. In contrast to CUT&Tag and related modifications[10,14–18], DynaTag utilises a physiological intracellular salt solution[19] throughout all nuclei handling steps to retain specific interactions while suppressing non-specific protein–DNA interactions. We extensively validate DynaTag for specificity and demonstrate its utility for studying TF occupancy and their dynamics at single-cell resolution and in bulk, with limited sample sizes, using models highly relevant to stem cell biology and cancer research.

## Results

### Robust genome-wide mapping of transcription factor-DNA interactions under physiological salt conditions

We rationalised that physiological intracellular salt conditions are required to preserve TF–DNA interactions in situ, as supported by our previous studies using an intracellular salt buffer to maintain the correct topology of dynamic nucleic acid structures inside living cells[19]. For this reason, we generated the DynaTag physiological salt buffer containing 110 mM KCl, 10 mM NaCl and 1 mM MgCl$_2$. This cation buffer composition is based on electrophysiological salt concentration measurements in situ[20] and thereby ensures the retainment of specific TF–DNA interactions during sample preparation (Fig. 1A). To first validate DynaTag alongside CUT&Tag, we profiled TFs and histone modifications involved in pluripotency and differentiation in mouse embryonic stem cells (ESCs). We included targets (CTCF, H3K4me3, and H3K27me3) that can be efficiently mapped by CUT&Tag due to their high DNA affinity and resistance to high-salt conditions during nuclei sample handling[10]. In addition, we included OCT4, SOX2, NANOG, and MYC, TFs associated with ESC identity, as well as YAP1. Only DynaTag but not CUT&Tag successfully generated sequencing libraries for all TFs (Supplementary Fig. 1A). Importantly, negative controls (using no primary but only secondary antibodies and pA-Tn5) from both techniques did not result in libraries. To better investigate which component of the DynaTag buffer was crucial for retaining specific TF–DNA interactions, we performed a systematic comparison of the DynaTag physiological salt buffer with other buffers used in CUT&Tag-adapted protocols. We utilised the nuclei washing procedure described in ACT-seq[17], CoBATCH[14], and G-quadruplex CUT&Tag[18]. Our results show that the wash buffers used in ACT-seq and CoBATCH caused a dramatic loss in library yield and led to untargeted Tn5 tagmentation in ACT-seq IgG control libraries (Supplementary Fig. 1B, C). Similarly, the use of reduced NaCl concentrations (121 or 150 mM) in the CUT&Tag wash buffer also led to untargeted tagmentation in the IgG controls (Supplementary Fig. 1D, E). Moreover, we showed that the 300 mM KCl buffer used in G-quadruplex CUT&Tag was responsible for failed library generation for all five tested TFs (Supplementary Fig. 1F). Consistent with this result, increasing the KCl concentration from 110 to 300 mM in the DynaTag buffer also abolished library generation (Supplementary Fig. 1G). To further exclude the occurrence of untargeted tagmentation, we performed DynaTag in the presence or absence of MgCl$_2$ in the wash buffer, in both fixed and native nuclei. Our results showed that neither the absence of MgCl$_2$ nor the fixation status affected

library yields for any of the tested TFs or the IgG controls (Supplementary Fig. 1H–J).

Omni-ATAC-seq uses Tn5 to map accessible chromatin by untargeted tagmentation[21]. We anticipated that replacing the Omni-ATAC-seq tagmentation buffer with the DynaTag buffer would eliminate library formation. We performed the Omni-ATAC-seq protocol in the absence of MgCl$_2$ or using buffers containing the same excess of pA-Tn5 but with either 150 mM NaCl, 300 mM NaCl, or the DynaTag buffer. Our results showed that only 150 mM NaCl generated libraries, suggesting untargeted tagmentation. In contrast, 300 mM NaCl or the DynaTag buffer did not produce any libraries (Supplementary Fig. 1K), implying a lack of untargeted tagmentation. Next, we assessed the quality of the DynaTag data, including reproducibility, peak quality and potential cell-cycle contributions. For this purpose, we stained the nuclei with DAPI post-tagmentation and sorted 10,000 nuclei in replicates, according to their cell-cycle state into G0/G1, S and G2/M phases. Genome-wide read distributions correlated well among the replicates (Fig. 1B) with excellent fraction of reads in peaks (FRiP) scores for all five TFs across the cell-cycle states (Fig. 1C). Of note, both DynaTag and CUT&Tag showed high similarity and high peak quality for CTCF and the histone marks when assessing genome-wide correlations and the FRiP scores (Supplementary Fig. 1L, M).

To identify peaks with unique or shared co-occupancy patterns, we systematically performed differential occupancy analysis (DOA) of the TFs (OCT4, SOX2, NANOG, MYC, YAP1) profiled in ESCs using DynaTag. Additionally, we examined whether the DynaTag peaks were located in accessible chromatin regions of ESCs, as determined by ATAC-seq. A genome browser snapshot illustrates two representative genes (Fig. 1D). The two highlighted genomic regions exhibited distinct occupancy patterns among the five TFs. In the left highlighted genomic region, which lies within the gene body of *Kdm3b*, all TFs except MYC showed clear occupancy signals. In contrast, the right highlighted region–located at the promoter of *Egr1*–displayed predominant occupancy by MYC (Fig. 1D). This specific snapshot demonstrated that TFs can independently occupy certain accessible genomic regions (Fig. 1D).

Overall, DOA identified six different sets of peaks exhibiting differential occupancies among the five TFs (Fig. 1E). For instance, MYC, YAP1, and OCT4 predominantly occupied specific genomic regions ($_{OSNY}$M, $_{OSN}$Y$_M$, O$_{SNYM}$), whereas other TFs showed significantly lower occupancy at these sites (FDR < 0.05, log2FC > 0.6); (Fig. 1E). In contrast, OCT4, SOX2, and NANOG co-occupied the same regions (OSNY$_M$, OSNY$_M$, $_O$SN$_{YM}$), either together with MYC or YAP1 or in the absence of MYC, YAP1, and OCT4 (Fig. 1E). These occupancy trends indicate specific regulatory programmes in which YAP1 and MYC act mutually exclusive together with OCT4, SOX2, and NANOG in ESCs, further underlining the specificity of DynaTag.

Analysis of DynaTag peaks from all five TFs at transcription start sites (TSS) of ChIP-seq derived target genes from ChIP-Atlas[22], revealed a remarkable coverage at the expected target genes for each TF (Supplementary Fig. 2). To compare the quality of DynaTag data with existing technologies used to map TFs in murine ESCs, we analysed matched publicly available datasets (OCT4, NANOG, and SOX2) generated by ChIP-seq and CUT&RUN. DynaTag displayed superior enrichment (signal-to-background) and resolution (sharper signal) of binding of all TFs at the TSS of the ChIP-seq derived target genes (Fig. 1F). Systematic analyses of known TF binding motifs in the DynaTag, ChIP-seq and CUT&RUN peak sets revealed that the expected motifs were among the top 10 enriched motifs (highlighted in red). Enrichment of expected motifs in DynaTag peaks was notably higher compared to matched CUT&RUN peaks (Fig. 1F), indicating superior specificity of DynaTag over CUT&RUN. Taken together, these results suggest that, in contrast to CUT&Tag and its modified protocols, DynaTag robustly maps TF–DNA interactions with high reproducibility across different cell cycle states and with excellent signal-to-

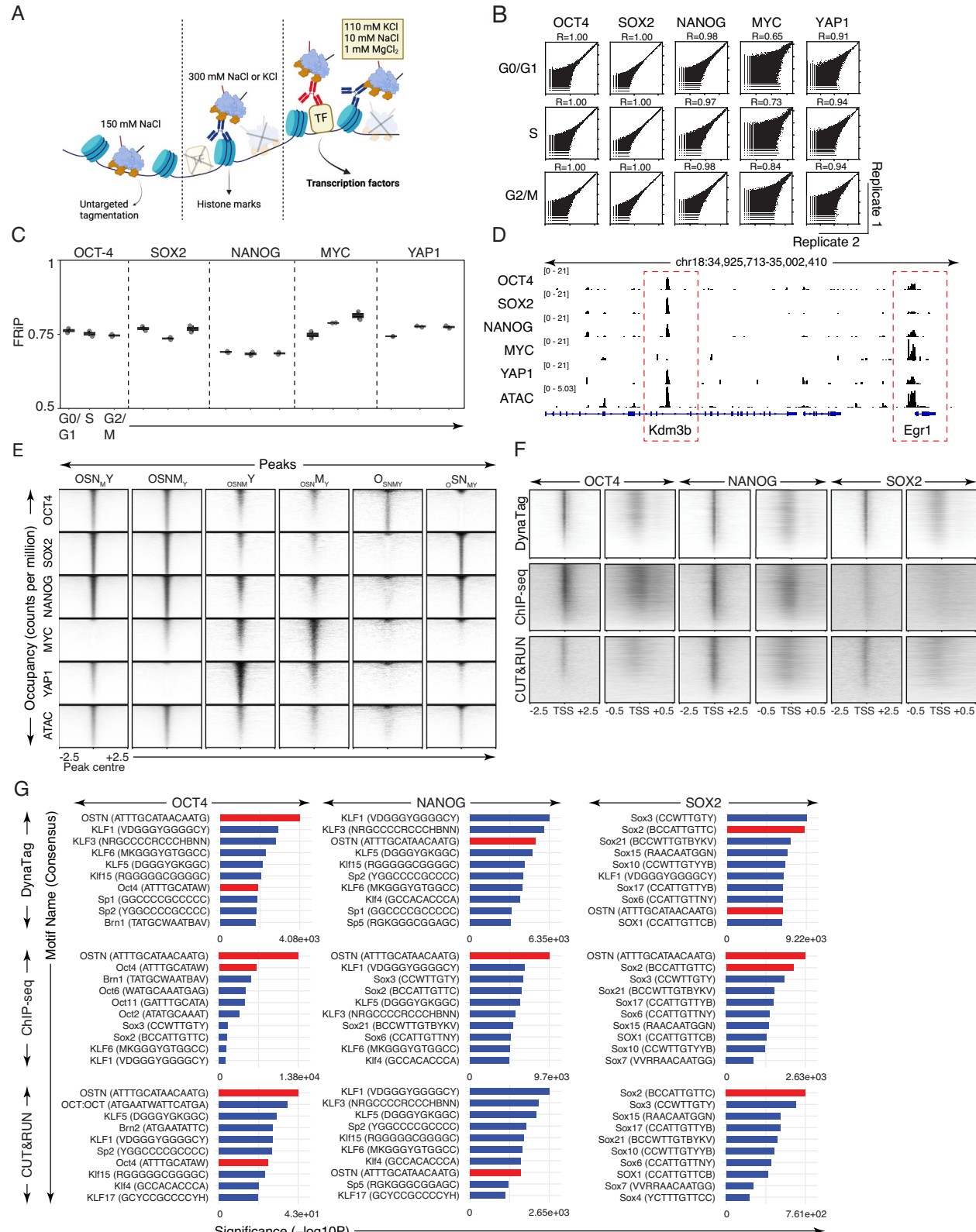

background ratios. Importantly, the physiological K⁺ concentration (110 mM KCl) in the DynaTag buffer is crucial to maintain the TF−DNA interactions during nuclei sample preparation, while preventing background signal from untargeted tagmentation. In comparison to CUT&RUN, DynaTag data showed superior resolution and specificity in uncovering the expected occupancy landscapes of all TF targets.

## Bulk DynaTag shows alterations in TFs occupancies between distinct cell states

To assess the ability of DynaTag to capture known TF alterations[23,24], we differentiated ESCs into epiblast-like cells (EpiLCs) and compared the occupancy of OCT4, SOX2, NANOG, MYC, and YAP1 in both cell states. First, we determined the FRiP scores for each cell state, which

**Fig. 1 | Establishment of DynaTag in mouse embryonic stem cells (mESCs).**
**A** Schematic illustration showing that only DynaTag captures TF interactions by using physiological salt concentrations (110 mM KCl, 10 mM NaCl, 1 mM MgCl₂) during nuclei wash steps. In contrast, the use of 150 mM NaCl or 300 mM NaCl or KCl lead to untargeted tagmentation or allows only for the targeting of histone marks, respectively. Created in BioRender. Hänsel–Hertsch, R. (2025) https://BioRender.com/o8b64lm. **B** Correlation between replicates of DynaTag-mapped TFs in ESC per cell cycle phase. **C** FRiP (Fraction of Reads in Peaks) of TFs in ESC per cell cycle phase (G0/G1, S, G2/M) called from DynaTag data. The box plot shows the medium (central line) with the 25th and 75th percentile (box). Each dot represents

an independent replicate $N = 2$. **D** IGV snapshots of TF read coverages profiled by DynaTag and ATAC-seq in ESCs. **E** Tornado plots of the DynaTag occupancies (OCT4, SOX2, NANOG, MYC, YAP1) in the six sets of regions OSNY$_M$ (23,420), OSN$_Y$M (28,681), $_{OSN}$Y$_M$ (2613), $_{OSNY}$M (1857), O$_{SNY}$M (1504), and $_O$SN$_{YM}$ (5207) revealed by DOA in ESCs. Occupancy is shown as counts per million for 2.5 kb up- and downstream of the peak centres. **F** Tornado plots of the DynaTag data and publicly available ChIP-seq and CUT&RUN data sets for OCT4, SOX2, and NANOG in ESCs. The signals are shown for 2.5 and 0.5 kb up- and downstream of the transcription start site (TSS). **G** Top 10 enriched motifs for each TF after motif enrichment analysis using HOMER2. Red bars show the expected motif for each TF.

were found to be equally good for all TFs and comparable between ESCs and EpiLCs (Supplementary Fig. 3A).

DOA of matched TF DynaTag datasets for each cell-cycle state (G0/G1, S, and G2/M) revealed a consistent and substantial decrease in NANOG occupancy across all cell cycle phases. For example, we found on average 15,681 peaks (1807 stdev., log2FC > 0.5, FDR < 0.05) that were significantly higher in read coverage in ESCs, while only 8869 peaks (2210 stdev., log2FC < −0.5, FDR < 0.05) showed a significantly higher read coverage in EpiLCs (Fig. 2A–C). In contrast to NANOG, OCT4, and SOX2 showed a similar number of regions with increased and decreased occupancy changes, while MYC and YAP1 exhibited an overall increase in occupancy in EpiLCs compared to ESCs (Fig. 2A–C). In line with our previous analysis showing specific TF co-occupancy behaviours in ESCs (Fig. 1D, E), we next investigated whether these behaviours could be quantitatively assessed between ESC and EpiLC states. We found that independent of increase or decrease of occupancy for any of the OSN (OCT4, SOX2, and NANOG) factors (Fig. 1E), the other TFs (O or S or N) showed a similar behaviour in co-occupancy (Fig. 2D). For example, for DynaTag peaks that showed a decrease in SOX2 occupancy in EpiLC vs ESC states (ESC > EpiLC), OCT4 and NANOG also displayed a decrease in occupancy in the same DNA regions, thus suggesting co-occupancy (Fig. 2D). In contrast, YAP1 and MYC exhibited distinct occupancy patterns without co-occupancy by the other TFs (Fig. 2D).

To independently validate the DOA results, we assessed the protein levels of these TFs in the chromatin-bound fraction of ESCs and EpiLCs by western blot analysis (Fig. 2E) and performed gene set enrichment analysis derived from RNA-seq data (Fig. 2F). These two independent assessments confirmed decreased and increased occupancies of NANOG and MYC, respectively, while confirming similar trends of OCT4 and SOX2 in EpiLC versus ESC states. Our findings of increased YAP1 occupancy in EpiLCs compared to ESCs are in line with a previous report demonstrating that YAP1 overexpression in ESCs promotes EpiLC formation[25].

Next, we aimed to investigate whether ATAC-seq TF footprint analysis would be sufficient to predict the altered TF activities between ESC and EpiLC states revealed by DynaTag (Fig. 2A–C). However, these predictions failed to accurately capture changes in TF occupancy (Fig. 2G). Finally, we retrieved ChIP-seq-derived peaks[22] for NANOG, OCT4, and SOX2 from ESC and EpiLC states, and repeated DOA with the DynaTag libraries using these ChIP peaks. These results revealed similar trends as observed for the DynaTag peaks, thus further validating previous DynaTag observations (Supplementary Fig. 3B–D). Collectively, these findings underscore the capability of DynaTag to robustly map TF dynamics in distinct cell states. Of note, TF footprint prediction is not robust enough to predict TF occupancy; instead, direct mapping using DynaTag of TF−DNA interactions is necessary to accurately delineate TF activities and their roles in gene regulation.

### Single-nuclei DynaTag reveals TF occupancy alterations in different cell states

CUT&Tag can robustly profile histone modifications at single-cell resolution[11], providing tagmentation-based methods the unique ability to characterise complex samples and multicellular organisms. To

robustly map TFs at the single-cell level, we established single-nuclei DynaTag (snDynaTag) and targeted NANOG, MYC, and OCT4 TFs in ESCs and EpiLCs. Genome browser examples show that the merged coverage (aggregated coverage) from single-nuclei profiles per TF and condition (ESC or EpiLC) aligns well with the respective bulk DynaTag profiles (Fig. 3A). The bottom row of the genome browser snapshots displays peaks where a clear change in occupancy occurred for the TFs, i.e., NANOG shows a loss of occupancy at particular peaks in the EpiLC state that are clearly visible in the ESC state. In contrast, the bottom examples for MYC and OCT4 show a clear gain of occupancy in the EpiLC compared to the ESC state. Importantly, FRiP scores and genome-wide correlation with the respective bulk data sets indicate that snDynaTag shows similar quality to and correlates well with the bulk DynaTag landscapes (Fig. 3B and Supplementary Fig. 4A). To address whether snDynaTag has the potential to uncover differential occupancy of a particular TF in two closely related cell states, we derived fold changes of the genome-wide coverages of the aggregated TF snDynaTag for ESC vs EpiLCs. Coverage analyses of fold changes in the combined peaks derived from the two aggregated snDynaTag data sets revealed thousands of peaks for which the TFs displayed increased (log2FC > 0.5) or decreased (log2FC < −0.5) binding (Fig. 3C, D). In accordance with bulk NANOG and MYC DynaTag DOA trends (Fig. 2A–C), snDynaTag showed an overall loss in NANOG occupancy while an increase for MYC occupancy in the EpiLC state was observed (Fig. 3C, D). We removed nuclei with low counts (2nd percentile; Supplementary Fig. 4B) and applied a $z$-score normalisation approach (see 'Methods') to account for single-nucleus coverage at peaks that showed consistent changes across nuclei in the ESC versus EpiLC groups (Fig. 3E). We next investigated whether dimensionality reduction analysis of individual TF profiles from ESC and EpiLC states would reveal clusters predominantly populated by one cell type over the other. For all three TFs, we revealed distinct clusters originating from TF occupancy in ESC or EpiLCs (Fig. 3F, G). This suggests that TF snDynaTag can resolve cell-type-specific occupancy of TFs, thus demonstrating the capability to uncover alterations in TF occupancy with single-cell resolution.

### Small cell lung cancer tumours display differential transcription factor occupancy in response to chemotherapy

To discover dynamic changes of TF−DNA interactions in a complex disease model in vivo, we investigated small cell lung cancer (SCLC). This cancer type harbours key alterations in TFs of the p53 family (e.g. *TP53* and *TP73*)[26–28], developmental and oncogenic TFs (e.g. NFIB and MYC)[28–31], along with the expression of key SCLC subtype-defining TFs (e.g. ASCL1, NEUROD1, POU2F3, and YAP1)[32]. Importantly, nearly all tumours respond to platinum-based therapy but then acquire chemoresistance[33]. For this reason, we aimed to investigate changes in TF occupancies following chemotherapy treatments in a patient-derived xenotransplantation (PDX) model for SCLC. The PDX model showed initially a good response to in vivo treatment with cisplatin and etoposide, and we subsequently analysed recurring tumours and respective control samples ($n = 2$ control, $n = 2$ chemo-treated). We employed bulk DynaTag and single-nuclei RNA-seq (snRNA-seq) to study chemotherapy treatments in a PDX model for SCLC, which

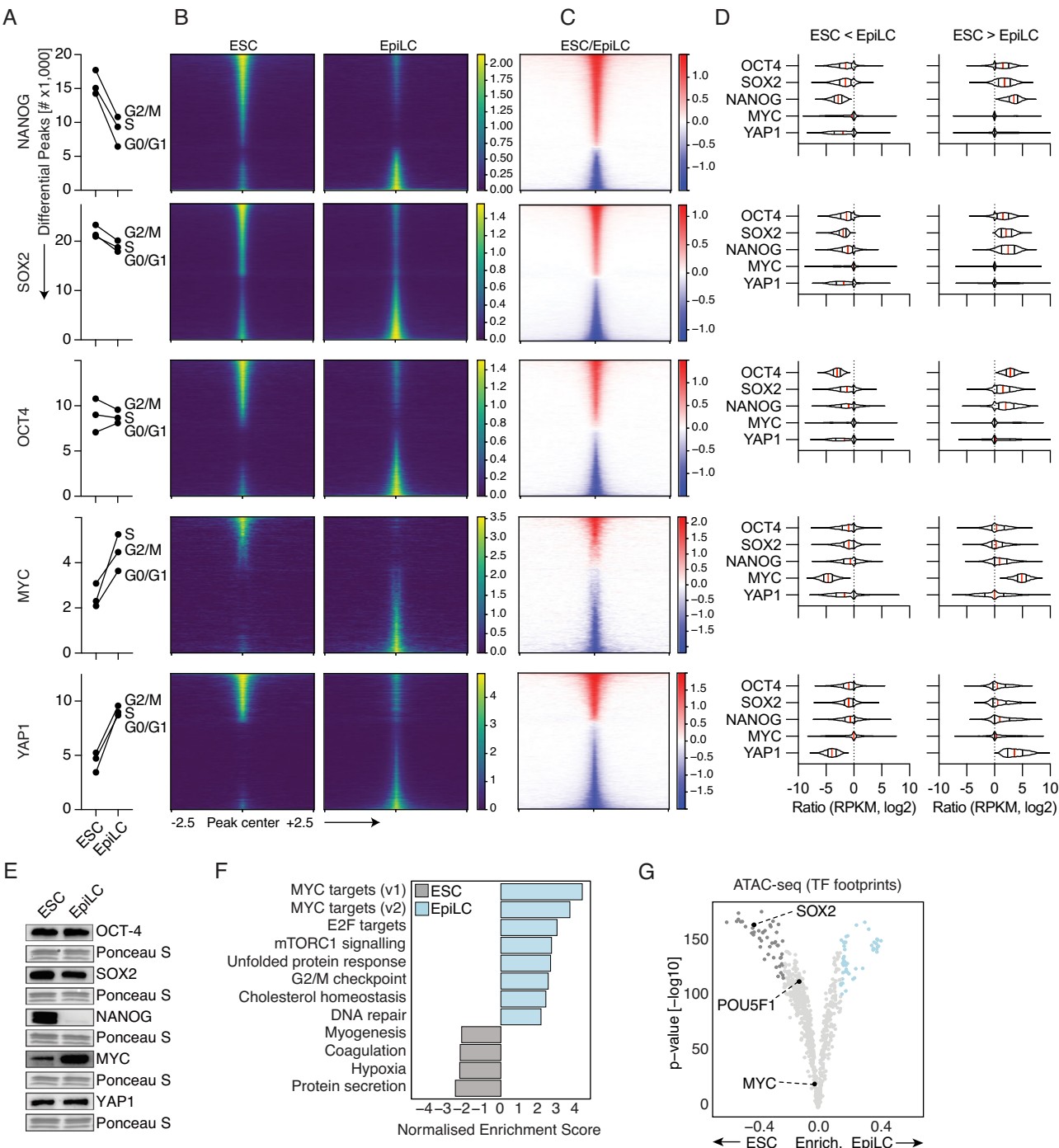

**Fig. 2 | Bulk DynaTag highlights TF dynamics in mESCs vs EpiLCs. A** Differential occupancy analysis (DOA) highlights the number of differentially occupied regions (edgeR) for each TF using the DynaTag peaks. $N = 2$. **B** Tornado plots of ESC and EpiLC TF bulk DynaTag occupancy show normalised (cpm) coverages (viridis) and ratios (log2, ESC vs EpiLC) of TF DynaTag at differentially bound (edgeR, >0.5 FC, FDR < 0.05) DynaTag peaks. **C** Summary of the number of differentially occupied regions (edgeR) for each TF. Red indicates increased, blue decreased occupancy of transcription factors in ESC vs EpiLC (ratio in cpm, log2). **D** Plots showing TF co-occupancies in EpiLC vs ESC states (ESC > EpiLC and EpiLC > ESC). **E** Western blot analysis of chromatin-bound proteins in ESCs and EpiLCs. Ponceau S serves as loading control. $N = 1$. **F** Normalised enrichment scores (NES) derived from GSEA of RNA-seq data of ESC and EpiLC. **G** Prediction of TF footprints by TOBIAS based on ATAC-seq data in ESC and EpiLC. P values were computed using the TOBIAS BIN-Detect module[51].

harboured mutations in *TP53* and *TP73*, and predominant expression of *ASCL1* (Fig. 4A and Supplementary Fig. 5)[26]. We mapped binding and quantified increased and decreased occupancies of 11 TFs (ASCL1, FOXA1, MYC, NEUROD1, NFIB, NRF1, POU2F3, p53, P73, SP2, and YAP1) in chemotherapy-treated relative to control tumours, for which transcripts were detectable (Fig. 4B and Supplementary Fig. 5A). We identified a substantial increase in FOXA1 and MYC, and a decrease in

ASCL1 and POU2F3 occupancies (Fig. 4B and Supplementary Fig. 6). These results suggest increased FOXA1 and MYC activity in SCLC cells following chemotherapy, and a decreased number of cells driven by ASCL1 and POU2F3 activity in vivo. To determine if changes in TF occupancy are linked to functional alterations in SCLC, we integrated DynaTag data with gene pathway activity changes from matched snRNA-seq data of the same tumours (Fig. 4 and Supplementary

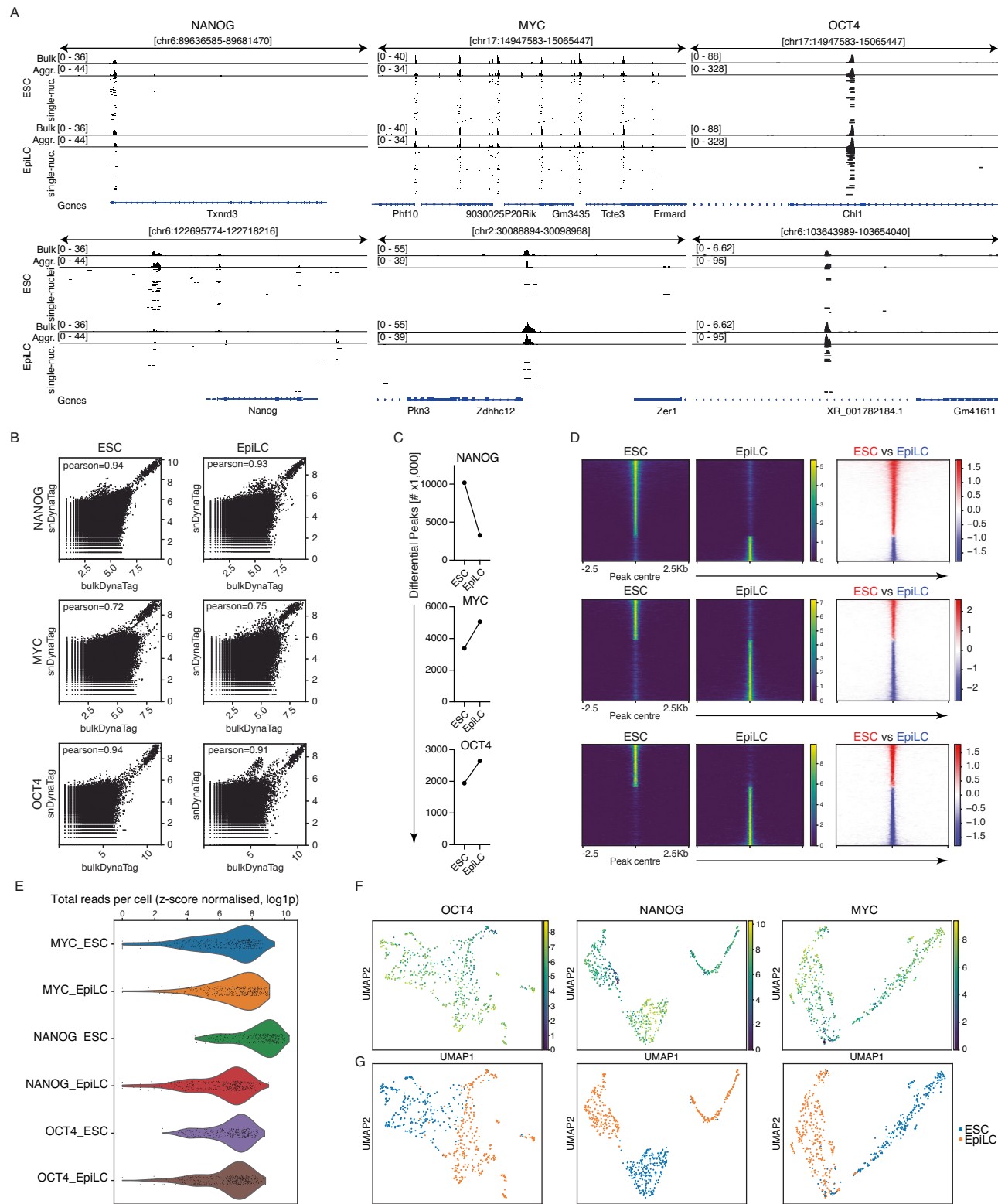

**Fig. 3 | TF dynamics at a single cell resolution. A** Genome tracks showing the comparison of bulk DynaTag, aggregate snDynaTag profiles and the occupancy from single nuclei for NANOG, MYC and OCT4 in ESC and EpiLC states. **B** Genome-wide correlation between bulk and snDynaTag data for each TF in ESC and EpiLC. **C** Number of differential occupied regions obtained for ESC vs EpiLC (log2, >0.5 and <−0.5) for each TF in the DynaTag peaks. $N = 1$. **D** Tornado plots of differential TF (NANOG, MYC and OCT4) snDynaTag occupied regions. Normalised aggregated

coverages are shown for ESC and EpiLC, as well as fold changes (log2, ESC vs EpiLC). **E** Distribution of log1p, or log(x + 1), values for the total number of reads in peaks that passed z-score filtering, shown for cells after quantile filtering and linked to each TF in ESCs and EpiLCs profiled by snDynaTag. **F** UMAPs of snDynaTag for OCT4, NANOG and MYC in ESC and EpiLC. Colour scale shows count signal. **G** UMAPs of snDynaTag for OCT4, NANOG and MYC showing distinct clusters for ESC and EpiLC states.

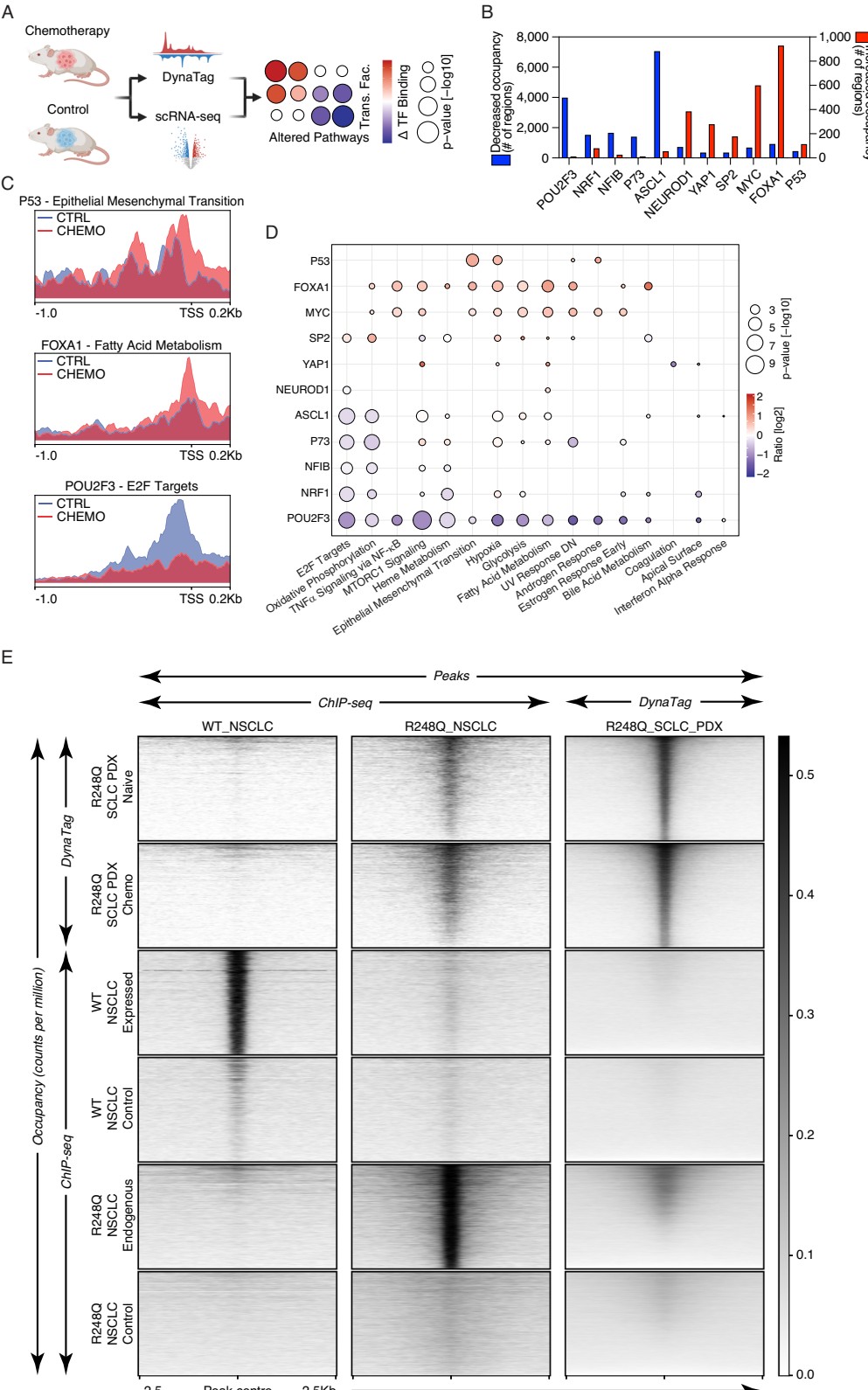

**Fig. 4 | TF dynamics of SCLC tumours in response to chemotherapy.**
**A** Illustration of functional genomics set-up to identify altered gene-pathways that are differentially occupied by TFs in a SCLC PDX model treated with and without chemotherapy. Created in BioRender. Hänsel−Hertsch, R. (2025) https://BioRender.com/urj5efd. **B** Number of differentially occupied regions of TFs. **C** Profiles of TF read coverages across gene promoters of specific hallmarks that are differentially enriched in transcriptomes derived from snRNA-seq. **D** Matrix displaying the ratio (log2) of TF signal (Chemotherapy/Control) at promoters of genes from the affected pathways. Size of bubbles represents the *p* value (−log10 transformed) derived from a paired *t* test per TF signal per pathway. Two-sided paired *t* test. **E** Tornado plot matrix of differential p53 occupancy for wild type (WT) and R248Q p53 mutant. Rows indicate the ChIP-seq or DynaTag signals, cancer type (SCLC PDX or NSCLC) and sample type (normalised read coverage (cpm)), which were plotted in peaks (WT_NSCLC (3034), R248Q_NSCLC (3875)) derived from the different ChIP-seq and DynaTag data sets (columns).

Figs. 5 and 6). Functional integration revealed a significant increase in FOXA1 and MYC occupancies, as well as a significant decrease in ASCL1 and POU2F3 occupancies at the promoters of genes identified as altered in the GSEA (Fig. 4C, D and Supplementary Fig. 7). Transcriptional plasticity of key neuroendocrine TFs has recently been observed. Specifically, decreased *ASCL1* along with increased *MYC* gene expression has been linked to a non-neuroendocrine state, thus suggesting selection of this phenotype in our PDX model[29,34,35]. Our results further showed significantly increased FOXA1 and MYC occupancies at genes involved in several metabolism pathways, such as fatty acid metabolism (Fig. 4C, D). Intriguingly, we also uncovered a significant increase in mutant p53 occupancy (p.R248Q variant) at promoters of genes related to epithelial-mesenchymal transition (EMT) (Fig. 4C, D). This suggests that the p53 variant acts as a potential driver of EMT, serving as a potential chemotherapy escape mechanism in SCLC. This finding aligns with previous reports linking the p53 R248Q variant to EMT, cell locomotion and increased aggressiveness[36,37]. We next asked whether the occupancy pattern of p53 R248Q identified by DynaTag in SCLC would resemble that observed by ChIP-seq for the same variant in non-small cell lung cancer (NSCLC)[38]. Strikingly, the DynaTag SCLC read coverage was notably enriched in peaks derived from the NSCLC p53 R248Q variant ChIP-seq data, while lacked any notable coverage in the wild type p53 peaks[39] (Fig. 4E). Vice versa, the NSCLC ChIP-seq coverage of the p53 R248Q variant was notably enriched in the DynaTag-derived peaks from SCLC but not in the peaks derived from wild-type p53 ChIP-seq data (Fig. 4E). Taken together, the DynaTag p53 landscapes acquired in SCLC suggest a gain-of-function for p53 R248Q in response to chemotherapy at promoters of signature genes related EMT, thus highlighting the potential of DynaTag for uncovering in vivo significant TF alterations.

## Discussion

CUT&Tag and its modified versions typically employ high-salt concentrations (e.g. 150–300 mM NaCl or 300 mM KCl) to prevent untargeted tagmentation by protein A-Tn5[10,14,15,17,18]. We reasoned that the intracellular environment, such as its ion composition and concentration, must be recapitulated during nuclei preparation steps to retain specific and prevent non-specific protein–DNA interactions. Our results show that continuously treating extracted nuclei with an intracellular-derived salt solution (DynaTag buffer) effectively suppresses untargeted tagmentation. Importantly, this post-extraction treatment also preserves specificity for both highly abundant, high-affinity binding targets (such as histone modifications) and low-abundance, low-affinity binding targets (such as TFs). We therefore conclude that conventionally used high salt concentrations reduce not only non-specific but also specific interactions, particularly for targets that are low in abundance and/or exhibit weak specific interactions with DNA. Of note, the physiological potassium concentration in the DynaTag buffer and the nuclei wash procedure were key to unlocking the robust mapping of TFs in small samples and at single-cell resolution. We emphasise that droplet-based single-cell CUT&Tag and nano-CUT&Tag rely on high-salt concentrations, which suggests that current protocols are likewise unsuitable as a robust method for mapping targets sensitive to stringent conditions. The authors themselves acknowledged that this remains a limitation requiring robust validation[15]. We have demonstrated here that DynaTag achieves superior data quality, occupancy resolution and signal-to-background ratio compared to ChIP-seq and CUT&RUN, as shown by comparisons of matched datasets generated from the same cellular system (Fig. 1F).

Overall, we successfully mapped 15 different TF targets in both simple cellular models and complex tissue samples, which are highly relevant for developmental biology and cancer research. Additionally, from a single PDX tumour sample, we successfully conducted 22 distinct DynaTag reactions covering 11 different TF targets, highlighting the quality, versatility and efficiency of this method. Given that many

researchers currently employ the original CUT&Tag procedure, we anticipate that transitioning to DynaTag will be both time- and cost-efficient.

Quantification of transcript and protein levels supported the differential occupancy patterns observed with DynaTag. However, predicted TF activities based on ATAC-seq footprinting did not match these occupancy trends between ESCs and EpiLCs. This discrepancy reinforces the importance of validating TF activity predictions derived from ATAC-seq data. Spatially resolved ATAC-seq and CUT&Tag allow localisation of epigenetic activities at the single-cell level within their native microenvironment[40–42]. We anticipate that spatially resolved mapping of TF occupancy will become feasible by integrating DynaTag with existing spatial CUT&Tag methodologies[40,43–45]. In addition, future work on development of spatial technologies for the mapping of TF-DNA interactions may provide a useful tool to uncover biologically relevant TFs alterations in situ in complex tissues.

In the discovery phase of our study, we examined a clinically relevant model of SCLC with high translational significance. SCLC remains among the deadliest cancers, with a 5-year survival rate below 5%. Although SCLC patients initially respond exceptionally well to chemotherapy, unknown resistance mechanisms lead to relapse in the majority of cases[33]. A multitude of TFs have been suggested to drive oncogenesis and neuroendocrine phenotypes in SCLC, which warrants the need to uncover TF occupancies throughout the deadly course of this disease. By applying DynaTag to identify previously unknown TF activities in SCLC following chemotherapy, we discovered significantly increased promoter occupancy by the gain-of-function R248Q variant of p53. This variant exhibited enhanced activity at promoters of genes relevant to EMT that also showed higher expression in chemotherapy-treated SCLC. Importantly, our findings provide insights into the role of the frequently mutated *TP53* locus in SCLC, particularly since point mutations in p53 are associated with sensitivity to chemotherapy treatment[26,27] and similar gain-of-function mutations have been documented in other cancer types[46,47]. Additionally, DynaTag revealed significantly increased occupancy of FOXA1 and MYC at promoters of genes related to multiple differentially expressed gene sets following chemotherapy, suggesting that these TFs may be critical for the survival of chemotherapy-resistant cells. Key changes in the experimental procedure allowed the development of DynaTag to resolve TF occupancy landscapes with so far unmatched sensitivity, selectivity and signal-to-background ratios in small samples and a complex disease model. Finally, we demonstrated that single-nuclei DynaTag of TFs is feasible, enabling future studies of TFs and other chromatin-associated factors in complex tissue samples.

## Methods
### Ethical statement
The patient sample S02730[26] was collected under approved protocols of the institutional review board (IRB, ID: 19-1164) of the University of Cologne and provided written informed consent to share potentially identifying individual-level data, gender: female, age at the time of first diagnosis: 64 years. All animal experiments were approved and conducted according to the regulations of local animal welfare authorities of the State Agency for Nature, Environment and Consumer Protection of North Rhine-Westphalia State of Nordrhein-Westfalen (LANUV, AZ: 84-02.04.2018.A002).

### Cell culture of murine embryonic stem cells (mESC) and epiblast-like cell (EpiLC) differentiation
Murine embryonic stem cells (kindly gifted by Dr. Simon Poepsel, University of Cologne) were cultured on gelatine-coated (Sigma-Aldrich, ES-006-B) dishes in 2i/L medium: (GMEM (Gibco, 11710035), 20% foetal bovine serum (Merck, ES-009-C), 1X NEAA (Gibco, 11140-035), 1X L-Glutamine (Gibco, 35035-038), 1000 ng/mL LIF (Gibco, A35934), 0.1 mM 2-mercaptoethanol (Gibco, 21985023), 1 mM sodium pyruvate (Gibco, 113060039), 3 μM CHIR99021 (AXON, 1386), 1 μM

PD0325901(AXON, 1408)) at 37 °C with 5% $CO_2$ and passaged every other day.

For epiblast-like cell differentiation, 200,000 ESCs were plated onto 15 µg/mL fibronectin-coated wells (Corning, 354,008) in 2i/L medium and allowed to attach overnight. The next morning, medium was changed to N2B27 medium as previously described[48] (0.5× DMEM/F-12 (Gibco, 10565-018), 0.5× Neurobasal Medium (Gibco, 21103049), 1× N2 (Gibco, 17502048), 1× B27 (Gibco, 17504044), 20 ng/mL activin A (STEMCELL, 78132.1), 12 ng/mL bFGF (STEMCELL, 78134), 1% KSR (Gibco, 10828010), 0.1 mM 2-mercaptoethanol (Gibco, 21985023), 0.04 ng/mL bovine serum albumin (Gibco, 15260037) for 48 h. N2B27 medium was changed after 24 h. Murine ESCs were routinely tested for mycoplasma contamination with MycoSpY Mycoplasma PCR Detection Kit (Biontex, M030-050).

## Antibodies and reagents
Primary antibodies used in this study are the following: from Merk Millipore: rabbit anti-CTCF (07-729); from Santa Cruz: mouse anti-POU2F3 (6D1) (sc-293402), mouse anti-YAP1 (sc-101199); from Abcam: rabbit anti-Histone H3 (tri methyl K4) (ab8580), rabbit anti-p73 (ab40658), mouse anti-p53 (ab1101), rabbit anti-MASH1 (ab74065), rabbit anti-SP2 (ab229468), rabbit anti-FOXA1 (ab23738), rabbit anti-c-MYC (ab32072), rabbit anti-OCT4 (ab181557); from Cell Signalling Technologies: rabbit anti-Tri-Methyl-Histone H3 (Lys27) (C36B11) (9733), rabbit anti-SOX2 (D9B8N) (23064), rabbit anti-NRF1 (D9K6P) (46743), rabbit anti-NANOG (D2A3) (8822), rabbit anti-NEUROD1 (D35G2) (4373); from Invitrogen: rabbit anti-NFIB (PA5-28299). Secondary antibodies are anti-rabbit IgG antibody (Novus Biologicals, NBP1 72763) or anti-mouse IgG antibody (Abcam, ab46540). A complete list of all chemical reagents is provided as Supplementary Data 1.

## Nuclei isolation from in vitro cultured cells
For nuclei isolation, cells were incubated in NE1 buffer (20 mM HEPES-KOH pH 7.9 (Carl Roth, 6763.3), 10 mM KCl (Sigma-Aldrich, P9333), 0.5 mM Spermidine (Sigma-Aldrich, S0266), 0.1% TritonX-100 (Sigma-Aldrich, X100), 20% glycerol (Carl Roth, 3783.2), 1× Complete Protease Inhibitor (Roche, 11836170001)) for 3 min on ice. Nuclei were fixed with 0.1% formaldehyde (Thermo Scientific, 28906) in PBS for 1 min at room temperature while gently agitated. Formaldehyde was quenched with 74 mM glycine (Sigma-Aldrich, 50046), and nuclei were pelleted by centrifugation at $450 \times g$ for 3 min at 4 °C.

## RNA isolation and RNA-seq from in vitro cultured cells
Total RNA from cultured cells were extracted using the NucleoSpin RNA Kit (Macherey-Nagel, 740955.50) according to the manufacturer's protocol. RNA-seq libraries were generated with the QuantSeq 3′ mRNA-Seq Library Prep Kit FWD with Unique Dual Indices for Illumina (Lexogen, 115.384) and sequenced on an Illumina NovaSeq 6000 platform using NovaSeq 6000 SP Reagent Kit v1.5 (100 cycles) (Illumina, 20028401). All RNA-seq experiments were conducted in four independent technical replicates.

## RT-qPCR
Total RNA from SCLC PDX tumours was extracted analogously to in vitro cultured cells. For cDNA synthesis, 1000 ng of total RNA was reverse transcribed using the High-Capacity cDNA Reverse Transcription Kit (Applied Biosystems, 4368814). Quantitative PCR was performed using pre-designed primers from Integrated DNA Technologies (IDT) on a Bio-Rad CFX Opus Real-Time PCR platform. No template controls served as background estimation. Data are shown as log10-transformed $2^{-\Delta Ct}$ values from two replicates.

## Chemotherapy treatment in a xenotransplant model for SCLC
Following previously described approaches[26,49], patient-derived tumour cells were isolated from the blood as circulating tumour cells and subcutaneously engrafted onto immunocompromised female (7–14 weeks) NOD scid gamma (NSG™) mice to establish a xenotransplant tumour model for patient S02730 (female, age at first diagnosis: 64 years). Following in vivo tumour growth and several passages in NSG female mice (7–14 weeks), tumour tissues were dissociated to single cell suspensions, and $1 \times 10^6$ cells were subcutaneously engrafted into both flanks of 7–14 weeks old NSG (NOD.Cg-Prkdcscid Il2rgtm1Wjl/SzJ) female mice. At a tumour volume of 100–200 mm³, mice were randomised into control and chemotherapy cohorts; $n = 2$ mice for each condition. Chemotherapy was applied intraperitoneally with 5 mg/kg cisplatin (day 1 and 8) and 10 mg/kg etoposide (days 1–3 and 8–10). Control animals received PBS as a vehicle control. Tumours were harvested at a maximum volume of 1500 mm³, as permitted by the animal welfare authorities. Animals were hosted in a pathogen-free facility on a 12 h light/dark cycle at uniform temperature and humidity.

## Nuclei isolation from snap-frozen tumour tissue
For nuclei isolation from frozen tissue, samples were briefly thawed on ice and minced on ice with a scalpel. Minced tissues were transferred into gentleMACS M Tubes (Miltenyi, 130-093-236) and 3 mL L1 (10 mM Tris-HCl pH 8.0 (Sigma-Aldrich, 93362), 5 mM CaCl2 (Sigma-Aldrich, C5670), 3 mM Magnesium acetate (Sigma-Aldrich, M2545), 2 mM EDTA (Sigma-Aldrich, E6758), 0.5 mM EGTA (Sigma-Aldrich, 3777), 1× Complete Protease Inhibitor (Roche, 11836170001), 1 mM DTT (Fisher Scientific, R0861), 0.1 mM PMSF (Sigma-Aldrich, 93482)) were added. Nuclei were isolated by running the 'Protein-M-Tube-1.0' programme. Three milliliters L2 (L1 with 0.4 % TritonX-100 (Sigma-Aldrich, X100) were added, and suspension was filtered through a 40 µm cell strainer (Sarstedt, 83.3945.40). Nuclei were pelleted by centrifugation at $450 \times g$ for 10 min at 4 °C with breaks at ~30%. Nuclei were resuspended in 1 mL L3 (L1 with 0.2% TritonX-100 (Sigma-Aldrich, X100)) and 3 mL sucrose buffer (1 M sucrose (Sigma-Aldrich, S7903), 10 mM Tris-HCl pH 8.0 (Sigma-Aldrich, 93362), 3 mM Magnesium acetate (Sigma-Aldrich, M2545)) were underlaid using a 5 mL syringe. Nuclei were pelleted by centrifugation at $450 \times g$ for 10 min at 4 °C with breaks at ~30%. Supernatant was carefully removed without agitating the layers, and nuclei were resuspended in the appropriate buffer for downstream application. For single-nuclei RNA-seq, all buffers were supplemented with 120 U/mL murine RNase inhibitor (NEB, M0314L).

## Single-nucleus RNA-seq of xenograft tumours
For single-nucleus RNA-seq, nuclei were isolated as described in the general methods. After isolation, nuclei were counted with trypan blue and fixed using the Evercode Nuclei Fixation v2 Kit (Parse Bioscience, ECF2003) and stored at −80 °C until library preparation. Library preparation of fixed nuclei was performed with the Evercode WT Kit (Parse Bioscience, ECW02130) according to the manufactures' protocol. Libraries were paired-end sequenced on an Illumina NovaSeq 6000 platform using NovaSeq 6000 SP Reagent Kit v1.5 (200 cycles) (Illumina, 20028401) with 5% PhiX Control V3 (Illumina, FC-110-3001). Sequencing fastq files were demultiplexed into single-nucleus data using the Split Pipe pipeline off Parse Bioscience (v1.0.6). Single-nucleus RNA-seq data were analysed in R using the Seurat (v 5.0.1) package. First, all four samples were pre-processed separately to perform quality control, which included filtering data for low-quality cells by removing the upper and lower 2% of the nuclei-based transcript coverage. Subsequently, data was further normalised using the sctransform (v 0.4.1) package to prepare samples for integration. All four samples were then read into R in parallel and merged into one SeuratObject, the features were reduced to the union, and variable features were identified. Upon dimensionality reduction via PCA, neighbours and clusters were identified and plotted into UMAP. Cell types were assigned to clusters manually based on the expression of lineage-specific marker genes. To exclude non-malignant tumour microenvironment cells, the SeuratObject was subsetted to the

small cell lung cancer clusters. The expression of the eleven TFs was plotted as FeaturePlot into UMAP. For differential expression analysis, the count data were aggregated into pseudobulk counts per sample and a combined transcripts-by-sample count matrix was generated. The combined count matrix was filtered for transcripts of a rowSums <10. DESeq2 was run with standard parameters and the control samples as reference. Significantly differentially expressed genes were identified (log2FoldChange > 0.5 or <−0.5 and padj < 0.1).

A slightly more relaxed adjusted $p$ value of 0.1 was chosen as the single-nucleus RNA-seq was only performed in duplicates and therefore offers limited statistical power. For gene set enrichment analysis, the transcripts were pre-ranked by multiplying the log2foldchange by the −log10 of the adjusted $p$-value and assigning the sign of the log2-foldchange to the value to capture the magnitude and significance of the differential expression. GSEA was run using the local GSEA software (v4.3.3). The normalised enrichment score of significantly enriched pathways (NOM $p$ val < 0.05) was plotted using ggplot2.

### Western blotting of subcellular fractionated lysates

Western blot analysis of subcellular fractions was performed as previously described in ref. 50. Briefly, cells were first lysed in E1 buffer (50 mM HEPES-KOH pH 7.5 (Carl Roth, 6763.3), 140 mM NaCl (Sigma-Aldrich, S5886), 1 mM EDTA (Sigma-Aldrich, E6758), 10% glycerol (Carl Roth, 3783.2), 0.5% NP-40 alternative (Sigma-Aldrich, 492016), 0.25% TritonX-100 (Sigma-Aldrich, X100), 1 mM DTT (Fisher Scientific, R0861), 1× Complete Protease Inhibitor (Roche, 11836170001)) to isolate the cytoplasmatic fraction. Subsequently, soluble nuclear proteins were extracted in E2 buffer (10 mM Tris pH 8.0 (Sigma-Aldrich, 93362), 200 mM NaCl (Sigma-Aldrich, S5886), 1 mM EDTA (Sigma-Aldrich, E6758), 0.5 mM EGTA (Sigma-Aldrich, 3777), 1 x Complete Proteinase Inhibitor (Roche, 11836170001)). Chromatin-bound proteins were isolated by incubation with ~500 U Benzonase (Sigma Aldrich, E1014) for 20 min at room temperature in E3 buffer (500 mM Tris pH 6.8 (Sigma-Aldrich, 93362), 500 mM NaCl (Sigma-Aldrich, S5886), 1× Complete Protease Inhibitor (Roche, 11836170001)). To increase the isolation efficiency of chromatin-bound proteins, chromatin was sonicated for five cycles (30 s ON, 30 s OFF) on a high setting of a Diagenode Bioruptor plus before Benzonase digest. Debris from all fractions was removed by centrifugation for 10 min at 16,000 × g and 4 °C. Protein concentration was assessed with the Qubit Protein Assay Kit (Invitrogen, Q33211). 10–20 μg of protein was separated in Mini-PROTEAN TGX Precast Protein Gels (Bio-Rad, 4561086) and transferred onto nitrocellulose membranes (Bio-Rad, 1704270). Total proteins were visualised by Ponceau S (Carl Roth, 5938.1) and served as a loading control. After blocking the membrane in 5% milk in TBS-T (20 mM Tris pH 7.0, 150 mM NaCl, 0.1% Tween 20), membranes were incubated 1:1000 diluted primary antibodies overnight at 4 °C. The next day, membranes were washed three times with TBS-T and incubated with 1:2500 diluted anti-rabbit-HRP secondary antibody (Invitrogen, 32460) and visualised with Clarity Max Western ECL Substrate (Bio-Rad, 1705062). All images were acquired on the Bio-Rad Chemi-Doc MP Imaging System.

### Assay for transposase-accessible chromatin using sequencing (ATAC-seq)

ATAC-seq experiments were performed as previously described in ref. 21. Briefly, 50,000 isolated nuclei were tagmented with 2.5 μL TDE1 enzyme (Illumina, 15027865) for 30 min at 37 °C. Upon completion of tagmentation, enzymatic reactions were immediately stopped by adding 3 volumes of ERC binging buffer from the MinElute Reaction Cleanup Kit (QIAGEN, 28204). Purified DNA was dual-indexed and amplified with Nextera XT Index Kit v2 (Illumina, FC-131-2001). Libraries were paired-end sequenced on an Illumina NextSeq 1000 platform using NextSeq P2 Reagent Kit (100 cycles) (Illumina, 20044468) with 2% PhiX Control V3 (Illumina, FC-110-3001). All ATAC-

seq experiments were performed in three to four independent replicates.

### Transcription factor occupancy prediction by investigation of ATAC-seq signal (TOBIAS)

TOBIAS[51] was used to identify TF footprints in ATAC-seq data. We investigated TF footprints in the consensus peak regions of ESC and EpiLC in their respective BAM files. The analysis was performed as described for TOBIAS. TF footprints for all vertebrates were downloaded from (https://jaspar.genereg.net).

### Protein A-Tn5 purification and transposase loading

Protein A-Tn5 was expressed and purified as previously described[10]. Briefly, T7 Express $lysY/I^q$ competent *E.coli* (NEB, C3013I) were transformed with the pTXB1-pA-Tn5 (Addgene, 124601) and cultured in LB Broth (Sigma, L3022) with 100 μg/mL ampicillin (Thermo Scientific Chemicals, J60977.14) for 4 h at 37 °C prior to inoculation of additional 1970 mL LB broth supplemented with 100 μg/mL ampicillin for expression. The 2 L culture was incubated at 37 °C until OD$_{600}$ reached 0.5–0.7 and was then cooled to 4 °C for 1 h. Expression was induced by adding 0.25 mM IPTG (Thermo Scientific, R0393) overnight at 18 °C. Bacteria were pelleted by centrifugation at 9500 × g at 4 °C for 10 min and stored at −80 °C until purification. Bacteria were lysed in HEGX buffer (20 mM HEPES, pH 7.2 (KOH) (Carl Roth, 6763.3), 1 M NaCl (Sigma-Aldrich, S5886), 1 mM EDTA (Sigma-Aldrich, E6758), 10% glycerol (Carl Roth, 3783.2), 0.2% TritonX-100 (Sigma-Aldrich, X100)) supplemented with 1× Complete Protease Inhibitor (Roche, 11836170001) using a Sonicator (BIOBLOCK SCIENTIFIC, Vibra-Cell 75043) for 10 cycles (45 s on/15 s off) with 50% amplitude on ice. Lysate was cleared by centrifugation at 16,000 × g at 4 °C for 30 min. In the meantime, 2.5 mL chitin resin (NEB, S6651S) was packed into Econo-Pac Chromatography Column (BIO-RAD, 732-1010) and washed with 20 mL HEGX buffer. The washed chitin resin was incubated overnight at 4 °C on a roller with the cleared lysate to bind protein A-Tn5. The next day, the resin was washed twice with 20 mL HEGX buffer. Protein A-Tn5 was eluted by adding 6 mL HEGX buffer supplemented with 100 mM DTT (Fisher Scientific, R0861) for 48 h at 4 °C on a roller. Eluate was dialysed against 4 L of 2x Tn5 dialysis buffer (100 mM HEPES, pH 7.2 (KOH) (Carl Roth, 6763.3), 0.2 M NaCl (Sigma-Aldrich, S5886), 0.2 mM EDTA (Sigma-Aldrich, E6758), 1.7 mM DTT (Fisher Scientific, R0861), 0.2% TritonX-100 (Sigma-Aldrich, X100), 20% glycerol (Carl Roth, 3783.2)) using a Slide-A-Lyzer 10 MWCO Dialysis Cassette (Thermo Scientific, 66380) for 24 h. Protein was concentrated using an Amicon Ultra Centrifugal Filter, MWCO 30 (Merck, UFC9030) to reach ~11 μM protein A-Tn5. Concentrated protein was mixed with equal volume glycerol (Carl Roth, 3783.2) and stored at −20 °C.

For transposase preparation, 16 μL of 100 μM annealed Tn5MEDS A and Tn5MEDS B oligonucleotides were mixed with 100 μL of Protein A-Tn5 glycerol stock and incubated for 50 min at 23 °C. The prepared transposase was stored at −20 °C.

### Cleavage under targets and tagmentation (CUT&Tag)

CUT&Tag experiments were performed as previously described with the following modifications[10]. Instead of immobilisation of nuclei on concanavalin A conjugated beads, nuclei were washed by centrifugation at 450 × g for 3 min at RT. 3 × 10$^5$ fixed nuclei were incubated with 1:100 diluted primary antibodies in 50 μL W150 buffer (20 mM HEPES pH 7.5 (KOH) (Carl Roth, 6763.3), 150 mM NaCl (Sigma-Aldrich, S5886), 2 mM EDTA (Sigma-Aldrich, E6758), 0.5 mM Spermidine ((Sigma-Aldrich, S0266), 0.01% (w/v) Digitonin (Carl Roth, 4005.1), 1% (w/v) BSA (Sigma-Aldrich, A7030)) on a nutating platform at 4 °C overnight. The next day, nuclei were washed once with 200 μL W150 buffer and incubated for 1 h at room temperature with 0.5 μg anti-rabbit IgG antibody (NOVUS BIOLOGICALS, NBP1 72763) in W150 buffer. After three wash steps with 200 μL W150 buffer, nuclei were incubated with

1:200 diluted protein A-Tn5 in W300 (W150 buffer with 300 mM NaCl) for 1 h at RT. Nuclei were washed three times with 200 μL W300 buffer, and Tn5 was subsequently activated by adding 10 mM MgCl₂ (Invitrogen, AM9530G) for 1 h at 37 °C. Upon tagmentation, nuclei were counted using an automated cell counter and equal number of nuclei were transferred into new tubes and denatured in 50 μL Denaturation buffer (10 mM Tris pH 8.0 (HCl) (Sigma-Aldrich, 93362), 50 mM NaCl (Sigma-Aldrich, S5886), 0.2% (w/v) SDS (Fisher Scientific, BP8200100), 0.5 ng/mL proteinase K (Ambion, AM2546)). Denaturation was stopped by purification of the tagmented DNA fragments using the MinElute Reaction Cleanup Kit (QIAGEN, 28204). DNA was eluted in 23 μL EB buffer and amplified using previously described custom Nextera primers[10] and the NEBNext 2X PCR Mix (NEB, M0541L) and the following PCR programme: 58 °C for 5 min, 72 °C for 5 min, 98 °C for 30 s, 98 °C for 10 s, 60 °C for 10 s, repeat from step 4 for 14 cycles, 72 °C for 1 min. Amplified sequencing libraries were purified with 1.3× SPRI beads (MAGBIO, AC-60050) ratio, followed by elution in 26 μL EB buffer. Libraries were paired-end sequenced on an Illumina NextSeq 1000 platform using NextSeq P2 Reagent Kit (100 cycles) (Illumina, 20044468) with 2% PhiX Control V3 (Illumina, FC-110-3001). All CUT&Tag experiments were performed in two independent replicates.

## DynaTag
For DynaTag, $3 \times 10^5$ fixed nuclei were incubated with 1:100 diluted primary antibodies on a nutating platform at 4 °C overnight in AB buffer (25 mM HEPES pH 7.5 (KOH) (Carl Roth, 6763.3), 110 mM KCl (Sigma-Aldrich, P9333), 10 mM NaCl (Sigma-Aldrich, S5886), 1 mM MgCl₂ (Invitrogen, AM9530G), 0.01% (w/v) Digitonin (Carl Roth, 4005.1), 1% (w/v) BSA (Sigma-Aldrich, A7030)), 0.5 mM spermidine (Sigma-Aldrich, S0266)) in 0.2 mL PCR tubes. The next day, nuclei were pelleted by centrifugation at $450 \times g$ for 3 min at RT incubated for 1 h at room temperature with 0.5 μg anti-rabbit secondary antibody (NOVUS BIOLOGICALS, NBP1 72763) or 0.5 μg anti-mouse secondary antibody (abcam, ab46540). Nuclei were washed twice by gentle resuspension in 200 μL AB buffer. Nuclei were incubated with 1:200 diluted pA-Tn5 complex in AB buffer for 1 h at RT with gentle agitation. Nuclei were washed twice by gentle resuspension in 200 μL AB buffer. Tagmentation was initiated by incubating nuclei at 37 °C for 1 h in AB buffer supplemented with 10 mM MgCl₂ (Invitrogen, AM9530G). Upon tagmentation, nuclei were pelleted by centrifugation at $450 \times g$ for 3 min at RT and resuspended in 100 μL DynaTag buffer (AB buffer without BSA, spermidine and digitonin) and an aliquot was mixed with trypan blue and counted using an automated cell counter. Alternatively, nuclei can be separated by their cell cycle phase using FACS upon DAPI staining. 10,000 nuclei were transferred into new 0.2 mL PCR tubes and incubated for 3 h at 58 °C in 50 μL Denaturation buffer (10 mM Tris pH 8.0 (HCl) (Sigma-Aldrich, 93362), 50 mM NaCl (Sigma-Aldrich, S5886), 0.2% (w/v) SDS (Fisher Scientific, BP8200100), 0.5 ng/mL proteinase K (ambion, AM2546)). Denaturation was stopped by purification of the tagmented DNA fragments using the MinElute Reaction Cleanup Kit (QIAGEN, 28204). DNA was eluted in 23 μL EB buffer and amplified using previously described custom Nextera primers[10] and the NEBNext 2× PCR Mix (NEB, M0541L) and the following PCR programme: 58 °C for 5 min, 72 °C for 5 min, 98 °C for 30 s, 98 °C for 10 s, 60 °C for 10 s, repeat from step 4 for 12–20 cycles, 72 °C for 1 min. An appropriate number of cycles can be estimated via qPCR and will depend on epitope abundancy. Amplified sequencing libraries were double-sided size-selected using first a 0.5× SPRI beads (MAGBIO, AC-60050) ratio to remove large fragments followed by purification with 1.3× SPRI beads ratio, followed by elution in 26 μL EB buffer. All DynaTag experiments were performed in two independent biological replicates. Two control and two chemo-treated SCLC PDX samples from patient S02730 were used.

## Correlation plots between CUT&Tag and DynaTag
BAM files were prepared using deeptools (v3.5.1) multiBamSummary (bins -bs 10). Correlation was plotted via deeptools plotCorrelation (--corrMethod pearson --skipZeros --whatToPlot scatterplot --log1p).

## Integrative genome viewer (IGV) snapshots
Indexed BAM files were converted into cpm-normalised bigwig files using deeptools (v3.5.1) bamCoverage (-bs 10 --skipNAs --centerReads --normalizeUsing CPM -of bigwig). Bigwigs were loaded into IGV and visualised as group auto-scaled bars.

## MACS2 peak calling
Aligned BAM files were down-sampled to equal reads per TF using samtools view (-@ 8 -c -F 260) and sorted with samtools sort. Peaks were called from sorted BAM files via MACS2 (v2.2.7.1) (-f BAMPE -g mm --keep-all --nomodel --extsize 55 -B –SPMR) from mouse cells or (-f BAMPE -g hg --keep-all --nomodel --extsize 55 -B --SPMR) from human PDX samples and filtered for conserved peaks between replicates using bedtools multiIntersectBed. Conserved peaks per condition were concatenated, followed by sorting and merging to remove redundant peaks with bedtools sortBed | mergeBed. To count reads, bedtools coverage was used to generate bedgraph files, which were then converted into plain text files.

## Calculation of fraction of reads in peaks (FRiP)
For FRiP calculation, overlapping reads with peaks were identified using bedtools (v3.5.1) intersect (-wa -u), and reads were counted using samtools (v1.13) view (-c -F 0 × 2). Finally, overlapping peaks were divided by total reads. FRiP scores were plotted in R using ggplot2 (v3.5.0).

## Enrichment of ChIP-seq target genes
Target genes (±1 kb) per TF were downloaded from the ChIP-Atlas repository[22] and used to extract the chromosome coordinates to generate a BED file using the GenomicRanges (v1.54.1) R package. Random non-target genes were extracted via setdiff command. Overlap of called peaks and (non)-target genes was counted with countOverlap and plotted using ggplot2 (v3.5.0).

## Motif enrichment via HOMER2
HOMER2 (v5.1, 7-16-2024) was used to quantify the top 10 most enriched known motifs in the matched ESC DynaTag, ChIP-seq and CUT&RUN data. Prior to motif analysis, the peak files were (i) normalised to the same average peak size, (ii) processed using the Homer2 background function to calculate the background sequences underlying the peaks, and (iii) analysed for motif enrichment. For details, please refer to the code available at https://github.com/HaenselHertschEpiLab/DynaTag.

## Differential binding analysis of DynaTag derived from in vitro cultured cells
For differential binding analysis, peak count matrices were read into R and analysed with edgeR (v 4.0. 16). Count matrices were pre-processed by removing low count peaks (rowSums <10). Dispersion was estimated with estimateGLMCommonDisp and estimateGLMTagwiseDisp, and data was fitted with glmFit followed by statistical estimation using glmLRT. Significant peaks (FDR <0.05 and logFC >0.5, and logFC <−0.5) were identified and plotted using ggplot2 (v3.5.0).

## Differential binding analysis of PDX-derived DynaTag data
DiffBind (Galaxy Version 2.10.0) coupled with DESeq2 and the significance threshold FDR < 0.1 was used in usegalaxy.eu[52] to identify differentially occupied regions. Default median-of-ratios was used to normalise the count matrix by DEseq2.

## Single-nuclei DynaTag

Around $4 \times 10^5$ mESCs were seeded for this experiment. Cells were harvested and kept on ice, and nuclei were isolated and fixed as described above.

Approximately $300 \times 10^5$ nuclei were incubated on a rotator at 4 °C overnight with primary antibodies at a 1:50 dilution in 100 μL AB buffer (25 mM HEPES pH 7.5 (KOH), 110 mM KCl, 10 mM NaCl, 1 mM MgCl$_2$, 0.01% (w/v) Digitonin, 1% (w/v) BSA, 0.5 mM spermidine).

Nuclei were centrifuged at $450 \times g$ for 10 min and incubated with anti-Rabbit or anti-Mouse IgG antibody (1:50 dilution) on a rotator at RT for 1 h. Nuclei were then washed twice with 200 μL AB buffer and incubated with a 1:100 dilution of pA-Tn5 adaptor complex on a rotator at RT for 1 h. Cells were washed twice with 200 μL AB buffer, resuspended in 100 μL Tagmentation buffer (AB buffer with 10 mM MgCl$_2$) and incubated at 37 °C for 1 h. After tagmentation, nuclei were pelleted at $600 \times g$ for 3 min and resuspended in 150 μL DynaTag buffer (25 mM HEPES pH 7.5 (KOH), 110 mM KCl, 10 mM NaCl, 1 mM MgCl$_2$).

For single-cell DynaTag, the ICELL8 cx nanoliter dispensing system (TaKaRa) was used similarly as reported for CUT&Tag[10]. Nuclei were counted using trypan blue to 1.4 cells/20 nL in PBS and 1× Second Diluent (Takara Bio USA, Cat. 640196). Nuclei were stained with DAPI for visualisation, and 80 μL/sample was loaded to a source loading plate. Control wells with PBS (80 μL) were also included in the source plate. Nuclei were dispensed onto an ICELL8 350 v chip at 20 nL per well twice. After each dispense, chips were sealed, centrifuged at $3220 \times g$ for 3 min and scanned using the automated microscopy image analysis software (CellSelect, Takara Bio USA). Additional single nuclei were manually selected using the manual triage function, and a filter file containing information of single-nuclei containing wells and control wells was generated. We obtained ~1500 single nuclei per chip. Then, 35 nL denaturation buffer (10 mM Tris pH 8.0, 50 mM NaCl, 0.2% SDS, 0.5 ng/mL proteinase K) was dispensed onto the single-nuclei containing wells, and chips were incubated in a thermocycler at 58 °C (80 °C lid) for 3 h followed by incubation overnight at room temperature. Next, chips were centrifuged at $3220 \times g$ for 3 min and 35 nL quenching buffer (10 mM Tris pH 8.0, 0.1 mM phenylmethanesulfonyl fluoride (PMSF), 2.5% Tween 20) was dispensed.

For indexing, 35 nL of 72 i5 and 72 i7 (set A and set B) unique indexes were dispensed on the single nuclei-containing wells. After the dispense, chips were sealed, centrifuged at $3220 \times g$ for 3 min and on-chip PCR was performed using the KAPA HiFi PCR kit (Roche) (2,7x HIFI fidelity buffer with Mg stock, 0.05 U/μL HiFi, 0.8 mM dNTPs) and the following programme: 58 °C for 5 min, 72 °C for 10 min, 98 °C for 2 min, 98 °C for 15 s, 58 °C for 30 s, 72 °C for 10 s, repeat steps 20 times from step 4, final elongation at 72 °C for 1 min.

PCR products were collected by centrifugation at $3220 \times g$ for 10 min at 4 °C using the single-cell collection kit (TaKaRa, 640212). Libraries were size-selected using SPRI beads. First, a ratio of 0.25× beads was used to remove larger fragments. Then, samples were transferred to a new tube, and a 1.3× ratio of beads was used. Beads were washed twice with 80% ethanol, let air-dry at RT for 5 min and incubated in elution buffer (10 mM Tris-Cl, pH 8.5) at RT for 10 min. Purified libraries were transferred to a new tube and quantified using Qubit and bioanalyzer.

Libraries were paired-end sequenced on an Illumina NextSeq1000 platform using NextSeq 1000/2000 P2 Reagents (100 Cycles) v3 (100 cycles)(Illumina, 20046811) with 0.5% PhiX Control V3 (Illumina, FC-110-3001).

All CUT&Tag experiments were performed in two to three independent replicates. For analysis, BCL files were demultiplexed into separate fastq files per cell using cellranger (v7.0.1). Fastq files were aligned to mm39 and mm10 using bowtie2 (v2.4.1) and aggregated into merged BAM files per TF using samtools (v1.13) merge for peak calling via MACS2 (v2.2.7.1). Using the aggregate called peaks, bedgraph files were generate per cell. Bedgraph files were converted into a large peak-by-cell count matrix and analysed in R using the Seurat (v 5.0.1) package. Using Seurat, the data was normalised like RNA-seq data and PCA followed by UMAP was run for dimensionality reduction.

## Calculation of the z-score for the peak locations

In order to correct for the variability of every peak signal across the single cells, we compute z-scores between the two cellular states *ESC* and *EpiLC*. Please note that the calculations below are, for sake of simplicity, only shown for one peak region. Since peaks are independent, the same mathematical expressions are valid for any location. To calculate the z-scores, we derive the variance $\sigma^2$ of the log-transformed location-averaged and cell-summed peak signal $\bar{z}$ by:

$$\bar{z} = \log_2\left(\frac{1}{n_b}\sum_{i=1}^{n}\sum_{j=1}^{n_b} c_{ij}\right) \tag{1}$$

$$\sigma^2 = \frac{1}{\ln(2)\, n_b} \cdot \frac{\sum_{k,l=1}^{n_b}\sum_{m=1}^{n}(c_{mk}-\hat{\mu}_k)(c_{ml}-\hat{\mu}_l)}{\sum_{i=1}^{n}\sum_{j=1}^{n_b} c_{ij}}, \tag{2}$$

where $n$ is the number of single cells, $n_b$ the number of bins around the peak location, $c_{ij}$ the $n \times n_b$ count matrix of the peak and cell average $\hat{\mu}_j = \frac{1}{n}\sum_{i=1}^{n} c_{ij}$. Let $c_{ij}^{ESC}$ be the $n^{ESC} \times n_b^{ESC}$ count matrix of the ESC cells and $c_{ij}^{EpiLC}$ the $n^{EpiLC} \times n_b^{EpiLC}$ count matrix of the EpiLC cells, we can then determine $\bar{z}_{ESC}$, $\sigma_{ESC}^2$ for ESC and $\bar{z}_{EpiLC}$, $\sigma_{EpiLC}^2$ for EpiLC by replacing the count matrix $c_{ij}$ with the respective cell-type-specific count matrix in the equations above. The z-score is then computed by:

$$z = \frac{\bar{z}_{ESC} - \bar{z}_{EpiLC}}{\sqrt{\sigma_{ESC}^2 + \sigma_{EpiLC}^2}}. \tag{3}$$

Peak area binning and z-score calculations

Narrow peak files were filtered for peak scores (column 5) equal to or higher than 1000. For each peak, the centre is extended by ±2.5 kb (kilobases). The extended peak region is then divided into 100 bins of 50 bases. To obtain the corresponding read counts for each bin per cell, the aligned BAM file for each cell is used as input for BEDTools/ 2.31.0-GCC-12.3.0. For all cells of each TF (MYC, NANOG and OCT4) and developmental condition (ESC and EpiLC), a merged count matrix was created for the binned peaks. To calculate the z-score, cells with more than 250 reads were selected, and read counts in each cell were scaled to 10,000 using the scanpy.pp.normalize_total() function. For each peak (containing 100 count bins), the z-score, $p$ value and adjusted $p$ value were calculated. Peaks with an adjusted $p$ value <0.001 were selected for downstream analysis and visualisation with UMAP.

## Analysis of the peaks and the single cells

Python's Scanpy package (version 1.10.4) was used to analyse the single-cell data. For each cell, the number of peaks with non-zero counts (non-zero peaks) was calculated. Cells falling within the lower 15% and upper 5% quantiles of non-zero peaks were removed (quantile filtering). For data quality control, violin plots were generated for each TF and developmental condition. These plots show the distribution of log $(x+1)$ values for: the total number of peak reads per cell, the number of non-zero peaks per cell, before and after quantile filtering, and the non-zero peaks selected by z-score filtering.

## UMAP visualisations

For data visualisation, the counts for all peaks were scaled to 10,000 using the scanpy.pp.normalize_total() function, and a log($x+1$) transformation was applied using scanpy.pp.log1p(). After z-score filtering, the dimensionality of the retained peaks was reduced to 10 principal

components using scanpy.pp.pca(). A neighbourhood graph was then computed with 25 neighbours using scanpy.pp.neighbours(). Finally, two-dimensional UMAP embeddings were generated using the scanpy.tl.umap() function.

### Single-nucleus RNA-seq and gene set enrichment analysis

Two control and two chemo-treated SCLC PDX samples from patient S02730 were used.

After isolation, nuclei were counted with trypan blue and fixed using the Evercode Nuclei Fixation v2 Kit (Parse Bioscience, ECF2003) and stored at −80 °C until library preparation. Library preparation of fixed nuclei was performed with the Evercode WT Kit (Parse Bioscience, ECW02130) according to manufacturer's protocol. Libraries were paired-end sequenced on an Illumina NovaSeq 6000 platform using NovaSeq 6000 SP Reagent Kit v1.5 (200 cycles) (Illumina, 20028401) with 5% PhiX Control V3 (Illumina, FC-110-3001). Sequencing fastq files were demultiplexed into single-nucleus data using the Split Pipe pipeline off Parse Bioscience (v1.0.6). Single-nucleus RNA-seq data were analysed in R using the Seurat (v 5.2.1) package. First, all four samples were pre-processed separately. Percentages of mitochondrial and ribosomal gene expression were computed for all cells. Cells with a ratio of mitochondrial versus endogenous genes expression higher than 0.1 were discarded. Additionally, cells expressing fewer than 1000 or more then 100,000 UMI counts, and fewer than 500 genes were also excluded. Raw expression data were normalised applying SCTransform function, regressing on percentage of mitochondrial gene expression. The top 3000 genes with the highest variance were computed. Principal component analysis (PCA) was performed using RunPCA function, and top 30 principal components were selected. All four samples were then integrated with batch correction by applying reciprocal PCA using the IntegrateLayers function. A k-nearest neighbours graph was computed and further refined into a shared nearest neighbour (SNN) graph using FindNeighbors function, using batch-corrected principal components as input. Cell clusters were defined at different resolutions applying Louvain algorithm to SNN graph using FindCluster function. To identify the resolution that produced the most stable set of clusters, we performed 20 subsamplings of the dataset, each time sampling 80% of the cells without replacement, and repeated the clustering procedure for all resolutions on each subsample. To assess cluster stability, we calculated Jaccard indices for each cluster at each resolution by comparing the original clusters with those obtained from each subsample. A cluster was considered stable if its median Jaccard index across subsamplings was higher than 0.75[53]. We finally selected the resolution that produced the higher number and fraction of stable clusters. The top 30 cluster-specific genes were identified by first applying PrepSCTFindMarkers function, followed by FindMarkers function with the following parameters: pseudocount.use = 0.2, assay = 'SCT', only.pos = TRUE, and min.pct = 0.2. Genes with an adjusted $p$ value < 0.01 were selected and ranked by log fold change (logFC); the top 30 genes were retained. For cell visualisation in two dimensions uniform manifold approximation and projection (UMAP) was computed with RunUMAP function. To exclude non-malignant tumour microenvironment cells, the SeuratObject was subsetted to the small cell lang cancer clusters. The expression of the eleven TFs was plotted as FeaturePlot into UMAP. For differential expression analysis, the count data were aggregated into pseudobulk counts per sample and a combined transcripts-by-sample count matrix was generated. The combined count matrix was filtered for transcripts of a rowSums <10. DESeq2 (v1.42.0) was run with standard parameters and the control samples as reference. Significantly differentially expressed genes were identified (log2FoldChange >0.5 or <−0.5 and padj <0.1).

A slightly more relaxed adjusted $p$ value of 0.1 was chosen as the single-nucleus RNA-seq was only performed in duplicates and therefore offers limited statistical power. For Gene Set Enrichment Analysis, the transcripts were pre-ranked by multiplying the log2-foldchange by the −log10 of the adjusted $p$ value and assigning the sign of the log2foldchange to the value to capture the magnitude and significance of the differential expression. GSEA was run using the local GSEA software (v4.3.3), the normalised enrichment score of significantly enriched pathways (NOM $p$ value < 0.05) was plotted using ggplot2 (v.3.5.0).

### Integration of bulk DynaTag with snRNA-seq data

For estimation of changes in occupancy of TFs via DynaTag at genes associated with altered pathways via GSEA (v4.3.3), the chromosomal coordinates of the TSS (+250; −1000 bp) of each enriched gene per pathway were extracted. Using these coordinated, bedgraph files were generated per TF per pathway from merged BAM files per factor per treatment condition. The counts in the bedgraph files were used to calculate the ratio of control to chemotherapy occupancy per gene followed by calculation of the mean per TF per pathway. To estimate the statistical significance, a paired $t$ test was performed per TF per pathway. The log2-transformed occupancy ratio was plotted as colour scale, and the −log10-transformed $p$ value was plotted as size parameter in a factor-by-pathway bubble plot matrix. The $y$-axis of the bubble plot was ordered according to the occupancy ratio for better readability. Of note, the $p$ values were not FDR-corrected to account for multiple test problems as the gene sets did not contain excesses numbers of genes. Moreover, the genes with a DynaTag count of 0 in one of the treatment conditions were excluded to prevent division by 0.

### Visualisation of DynaTag signal around the TSS of GSEA genes

To visualise the DynaTag signal at the genomic coordinates used to calculate the occupancy ratio, deeptools (v3.5.1) computeMatrix (reference-point --beforeRegionStartLength 1000 --afterRegionStartLength 250 --referencePoint TSS --missingDataAsZero --skipZeros) followed by deeptools (v3.5.1) plotprofile (--perGroup --colours blue red --plotType=fill) was used.

### Statistics and reproducibility

No statistical method was used to predetermine sample size. No data were excluded from the analyses. The experiments were not randomised. The Investigators were not blinded to allocation during experiments and outcome assessment.

### Statement on experimental measurements

Measurements were taken from all samples.

### Reporting summary

Further information on research design is available in the Nature Portfolio Reporting Summary linked to this article.

## Data availability

The murine DynaTag, CUT&Tag and RNA-seq data are available at NCBI GEO under accession code GSE273872. The DynaTag and snRNA-seq data related to the PDX model S02730 are available at the European Genome-Phenome Archive, under restricted access by the Data Access Committee University of Cologne, using the accession code EGAS50000001074. The data generated in this study, underlying the figures, are provided in the Source Data file.The following published ChIP-seq and CUT&RUN datasets were used for OCT4, SOX2 and NANOG:SRR10992265 [https://www.ncbi.nlm.nih.gov/sra/?term=SRR10992265]SRR10992267 [https://www.ncbi.nlm.nih.gov/sra/?term=SRR10992267]SRR10992266 [https://www.ncbi.nlm.nih.gov/sra/?term=SRR10992266]SRR23310250 [https://www.ncbi.nlm.nih.gov/sra/?term= SRR23310250]SRR23310248 [https://www.ncbi.nlm.nih.gov/sra/?term= SRR23310248]SRR23310227 [https://www.ncbi.nlm.nih.gov/sra/?term= SRR23310227] The following published ChIP-seq

datasets were used for p53:SRR12884191 [https://www.ncbi.nlm.nih.gov/sra/?term=SRR12884191]SRR12884190 [https://www.ncbi.nlm.nih.gov/sra/?term=SRR12884190]SRR27596890 [https://www.ncbi.nlm.nih.gov/sra/?term= SRR27596890]SRR27596891 [https://www.ncbi.nlm.nih.gov/sra/?term= SRR27596891] Source data are provided with this paper.

## Code availability
Computation code is available under https://github.com/HaenselHertschEpiLab/DynaTag and within Zenodo[54].

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

## Acknowledgements

This work was founded by the German Research Foundation (Deutsche Forschungsgemeinschaft, DFG)—SFB1399, project ID: 413326622— to R.H-H., R.K.T., M.P. and J.G, SFB1530 to M.P., and SFB1588 to M.P. Additional Funding was received through the Center of Molecular Medicine Cologne (R.H.H.), DFG SFB1399 (INST 216/1057-2), Fritz Thyssen Foundation (10.22.1.010MN), from the program "Netzwerke 2021", an initiative of the Ministry of Culture and Science of the State of North-Rhine-Westphalia for the CANTAR project to R.H.H., R.K.T, J.G. and M.P., by DFG HA 8562/4-1 and DFG RU5504 (HA 8562/5-1) to R.H.H., by the German Federal Ministry of Education and Research (BMBF, e:Med consortium InCa, grant 01ZX1901A and 01ZX2201A) to R.K.T, J.G. and M.P., by the DFG (project ID: 497777992 to J.G.), by the translational cancer program of the German Cancer Aid (Deutsche Krebshilfe, project TACTIC, grant number 70115201 to R.K.T.), by the Bruno-Helene-Joester foundation (J.G. and M.P,) and the Jean Uhrmacher Foundation (J.G.). We thank the Cologne Center for Genomics for the library preparation and sequencing of the bulk transcriptomes. We are grateful to Dr. Simon Poepsel at the CMMC for sharing murine ESCs and Dr. Filippo Beleggia at the Department of Translational Genomics, University of Cologne, for sharing the antibody against NFIB. We thank Dr. Benjamin Czech Nicholson (Cancer Research UK Cambridge Institute) for critically reading the manuscript and providing valuable comments.

## Author contributions

Conceptualization: P.H. and R.H.-H. Investigation: P.H., L.K., O.v.R., and G.P. Formal analysis: P.H., R.H.-H., N.H., and G.B. Resources: R.H.H., J.G., R.K.T., and M.P. Writing—original draft: P.H., R.H.-H., and G.P. Writing—Review & editing: P.H., R.H.-H., L.K., J.G., R.K.T., G.P., N.H., and M.P. Supervision: R.H.H.

## Funding

## Competing interests

R.K.T. is a founder of PearlRiver Bio (now part of Centessa), a shareholder of Centessa, founder and shareholder of Epiphanes Inc. and a consultant to PearlRiver Bio and Epiphanes Inc. R.K.T. has received research support from Roche. J.G. is consultant to DISCO Pharmaceuticals and received honoraria from MSD and Boehringer Ingelheim. The remaining authors declare no competing interests.
