## [Transparent Peer Review file · Nature Communications]

DynaTag for efficient mapping of transcription factors in low-input samples and at single-cell resolution

Corresponding Author: Dr Robert Haensel-Hertsch

Version 0:

Reviewer comments:

Reviewer #1

(Remarks to the Author)

The authors describe a minor modification of the popular CUT&Tag method for chromatin profiling of small samples and single cells, showing that it can be used for transcription factors. The high salt washes used with the standard CUT&Tag protocol can dislodge many TFs, and here a slightly modified buffer appears to suffice to prevent loss. Indeed the data presented do show improved recovery relative to CUT&Tag performed using the standard procedure, with improved efficiency. However, this brief article is missing the kinds of details that someone interested in trying or switching from the current CUT&Tag protocol will need to know. Moreover, the narrowness of the advance, and the fact that some of the protocol modifications have been published by others, makes it unsuitable for a general interest journal like Nature Communications. Much more evidence is needed, including controls that clarify just what is the basis of the improvement over such a widely used method that would explain the technical basis for the proposed improvement.

Major issues:

- 1) The rationale behind the change in buffer is not explained, nor are the various components tested individually. The change in alkali metal ion from 150 mM NaCl to mostly KCl is not really novel, as it's routinely used by several labs for CUT&Tag when profiling G-quadruplexes, which are stabilized by K⁺, but that isn't mentioned nor is the change justified by the authors. The original CUT&Tag protocol cited and used for comparison (Kaya-Okur 2019) has been used for one of the TFs in this study, Sox2, and the improvement here might have been entirely from lowering the ionic concentration and using Mg⁺⁺ during pA-Tn5 binding to increase tagmentation efficiency. But there's no clue in the manuscript just what is responsible for the improvement.
- 2) The reduction in salt during tagmentation to improve efficiency is well-known, for example CoBATCH and ACT-seq published in 2019 but not cited used similar ionic concentrations. Also, in PMID: 33191916 for bulk and 34250209 for single cells, low-salt tagmentation greatly improved detection of active histone modifications and RNA Polymerase II. Although this change in protocol was not applied to TFs, the basic strategy of reducing salt to increase tagmentation is not original.
- 3) The presence of 1 mM Mg⁺⁺ in the DynaTag buffer is puzzling, because even at room temperature, a 1 hour incubation should result in tagmentation. Perhaps that is what is happening, but the rate of Protein A binding to the antibody is fast enough to avoid open chromatin mapping. Controls are needed to make clear just what is happening.
- 4) Tagmentation at 10 mM MgCl₂ needs to be explained mechanistically, as this high concentration, especially in light of point 2 above, may have unintended effects. Again, it is hard to understand why exchanging most of the Na⁺ with K⁺ and why Mg⁺⁺ can give such improved results relative to standard CUT&Tag. Controls are needed.
- 5) The comparisons to CUT&Tag are not sufficiently documented, with just gel images in a supplement. Snapshot comparisons to ATAC-seq is insufficient, and there is the concern that tagmentation of open chromatin will be a problem for some TF antibodies.
- 6) Although this protocol is applied to lightly fixed nuclei, the cited paper used native nuclei and later versions used lightly fixed and whole cells and whether the protocol applies to these variations is unclear. This will be important if the new protocol is to be widely adopted.

7) For such a minor tweak of the CUT&Tag protocol, which is used in hundreds of laboratories and is the basis for multiple kits, calling this DynaTag is not justified. Nor is a new name in the authors' interest, considering that if adopted, the same reagents and kits will suffice, so presenting this as an easily implemented improvement rather than a new method means that investments already made by industry and their customers can be taken advantage of. CoBATCH and ACT-seq are examples of CUT&Tag-like methods that were never adopted, and it would be unfortunate if what looks to be an improvement in a popular method were to be ignored even though the modifications are so simple. Maybe describe it as CUT&Tag with DynaTag buffer instead.

Minor issues:

- 1) There have been many modifications to the original CUT&Tag protocol from 2019, which is referred to as the protocol used for comparison, making just a reference to the original paper insufficient.
- 2) The GEO entry number is unrecognized by GEO.
- 3) For such a brief methods paper, a step-by-step protocol would be helpful, otherwise it is unlikely to be adopted.

Reviewer #2

(Remarks to the Author)

The manuscript "DynaTag for efficient profiling of transcription factors in small samples and single cells" presents an approach for the improved profiling of protein-DNA interactions, particularly transcription factors (TFs). This is interesting, however this manuscript encompasses 4 major topics, each of which requires substantial additional analysis and documentation to affirm their claims.

First, the authors describe their desire to profile transcription factors by tagmentation, and describe this as "elusive". They show data on a panel of transcription factor antibodies that do not work in their hands by CUT&Tag, but do work with a modified protocol. The manuscript does not address the rationales for their modifications, or how these have improved profiling. This should be discussed in the text.

It is critical that these results be distinguished from general accessibility of genomic sites, which is not adequately demonstrated here. Looking at the details of their protocol modifications, the differences seem to be a different monovalent salt in many buffers, a lower concentration of monovalent salts, and the presence of magnesium in all buffers. It has been previously documented that conditions like these allow untargeted tagmentation (PMID: 32913232). Figure 1D attempts to address this with selected examples of factor profiling, but a more comprehensive analysis is needed to determine if and how much of profiling signal is explained by accessibility. Additionally, it is not clear that the presented ATAC data will be adequate; the shown examples have very small scales, and the other subpanels have widely different scales. The motif enrichment analysis presented in Figure 1E shows very poor enrichment of expected motifs (10X relative to random), suggesting that many of the called peaks actually lack high-quality motifs and are not expected binding sites. A more thorough analysis of signal in peaks compared to published profiling should also be included. Supplementary Figure 1D should also be expanded to consider signal. This section should also address the mechanism by which these buffer changes are acting. The authors attribute this to retaining factor-chromatin interactions during sample preparation, but this statement is unsupported.

The second topic covered here is a study of ESCs and differentiated epiblast-like cells. The changes in FRiPs and in UMAP counts is marginal between these two cell types, and it is not clear to me that there is really any difference. A more thorough analysis focusing signal at known peaks that change between these cell types for these factors is needed. I question whether the TF binding profiles should really be that different between these very similar cell types.

The third topic is a single-cell study of transcription factors in ESCs. This work is not adequately described or analyzed. It is standard in this field to report the quality of this data, including platforms, cell numbers and quality metrics, and these results should be included in the main. The supplementary data shows some evidence for cluster separation but with very poor distinction, raising questions about what these clusters mean. A previous study documented CUT&Tag transcription factor profiling in single cells, and should be referred to here (PMID: 33846645). This section should be substantially expanded.

The fourth topic is a study of small cell lung cancer xenograft and recurring tumors. This section lacks important documentation and analysis. It is not stated how many samples were analyzed, what their relationship is, how many cells, what is the quality of the data and results. Supplementary Figure 4D shows very poor segregation of multiple markers between the presumptive clusters, and an independent method of assigning or verifying cluster identity is needed. This section includes profiling of P53 which could be particularly of interest, and for this reason it should be assessed whether the antibodies used here for TFs detect the intended protein; in other words, do they give no peaks in cell types or settings without the target protein?

Minor comments:

The justification behind naming the approach "dynaTag" is unclear. It doesn't measure the dynamics of chromatin proteins, and previous publications have described the profiling of transcription factors. Perhaps consider a name that represents what about this protocol gives improved results.

The motivation behind comparing DynaTag to using ATAC-seqs alone to identify transcription factor binding is unclear. Previously ATAC-seq is used alongside motif analysis for estimated binding, Is this the motivation behind this hypothesis? Clarification would help with the flow of experiments.

Figure 1i is not labeled, and the legend describes it as "snDT", presumably scDT?

Previous publications have presented profiling of transcription factors by CUT&Tag, including the original publication which included profiling of CTCF, NPAT, and Sox2 in human cells, and other papers have presented profiling of Olig2 and of Rad21. Why do the authors think that these previous CUT&Tag experiments have worked while theirs do not?

Reviewer #3

(Remarks to the Author)

In the submitted manuscript, Hunold et al. describe a modified version of CUT&Tag, called DynaTag, with improved profiling of transient and dynamic DNA-binding transcription factors. Overall, the method and manuscript seem promising, yet not enough data is included with the manuscript to accurately assess the novel method.

Specific concerns:

1. Genome tracks depicting data generated via DynaTag should be shown in the main figures. The method is exclusively described by aggregative statistics over classes of peaks, with the exception of several small areas in figure 1D; example tracks of data generated by this method, over entire gene loci etc. should be shown to allow for comparison of the data to existing methods.
2. Related to point (1), the data for the scDynaTag as tracks for single cells should be shown. The data in figure 1I is not interpretable; the manuscript mentions the UMAP reveals "heterogeneity" within the population, but there are only two clusters in each UMAP and not enough data is presented to see that this heterogeneity represents biology and not technical artifacts. Can scDynaTag distinguish ESC from EpiLC, for instance?
3. Related to points (1) and (2), the data in figure 2C should be presented as tornado plots rather than aggregate traces over classes of loci, to allow for visualization of individual peaks.
4. More validation should be performed comparing DynaTag data with the ATAC-seq data; previous reports (PMID 32913232) have demonstrated that lower salt washes can lead to off-target pA-Tn5 insertions at accessible sites. Could the authors show something like tornado plots for their CUT&Tag vs. DynaTag vs. ATAC-seq data?
5. In figure 1E, what are the relative rankings of the discovered motifs? Do these motifs represent the most significant or well-represented motifs from each analysis?

Minor concerns:

1. More information regarding comparisons of DynaTag to CUT&RUN should be added to the manuscript. CUT&Tag is not efficient at capturing TF interactions, but CUT&RUN works well at profiling these interactions and is the more relevant comparator to DynaTag. The only point in the manuscript that mentions CUT&RUN is the last sentence which briefly mentions it needs more cells.
2. Figure 1A is not informative; elements describing the novel differences of DynaTag that allow for TF profiling should be added.
3. Throughout the manuscript, mouse proteins should have only the first letter capitalized, rather than all letters capitalized (which is the convention for human proteins).
4. Figure 1I is missing a letter label in the figure.
5. The color scales in figures 1I and S4 are too similar to clearly see differences.

Reviewer #4

(Remarks to the Author)

Version 1:

Reviewer comments:

Reviewer #1

(Remarks to the Author)

The authors have addressed all of my concerns, and I support publication in Nature Communications

Reviewer #2

(Remarks to the Author)

The revised manuscript "DynaTag profiles dynamics of transcription factor-DNA binding for low-input samples and at single-

cell resolution” has added new experiments and analysis that more thoroughly document the method. Supplementary Figure 1 fully documents the effects of the method differences here compared to CUT&Tag, ACT-se1, and CoBATCH, and will be useful for the field. Most of my concerns have been addressed; remaining comments are:

“For this reason, we generated the DynaTag physiological salt buffer containing 110 mM KCl, 10 mM NaCl and 1 mM MgCl₂. This cation buffer composition is based on electrophysiological salt concentration measurements *in situ*²⁰, and thereby ensures the retainment of specific TFDNA interactions during sample preparation (Fig. 1a).” This buffer seems to be the key innovation, but I’d like the authors to say more about this. It is not clear to me how this buffer would affect TF retention in formaldehyde-fixed nuclei. There is no data here that TFs are being lost from samples in other buffers, only that profiling works in the KCl buffer. The authors should consider literature describing that Na⁺ and K⁺ ions have different effects on chromatin compaction, and that this may affect antibody accessibility or tagmentation. This could be included in the Discussion.

There are issues with figures that should be clarified:

Figure 1: How many sites are represented in each box in Figure 1E. Is each box different numbers of sites? This is crucial to document how many sites overlap between the profiled factors. Is it expected that the Kdm3B site shown in Figure 1D would bind $\frac{4}{5}$ factors, including YAP1? What is the scale on the motif enrichment bar plots: the label says “-log₁₀P” but then is also labeled as exponents. The plots for all three methods should be on the same axis, so that they can be compared. For Oct4 in Figure 1F, it appears that only ChIP-seq recovers the expected motif with significance; the other methods do not meet standard cut-offs.

Figure 3E - log₁₀p is ln(1+x)?

Figure 4E - how many sites in each box? Is there no data for DynaTag profiling in wt?

DynaTag: Low-affinity binding is not the same as on-off dynamics, and the title misrepresents the method: “DynaTag profiles dynamics of...”; it does not measure dynamics, in the sense that live imaging or single-molecule footprinting do.

Reviewer #3

(Remarks to the Author)

The authors have addressed my concerns; I think the manuscript in its present form will be of interest to the field.

One minor suggestion; in figure 1, the "SOX2" label for the bottom left section of panel (F) is somewhat close to the arrow on the bottom X-axis of panel (E); this is slightly confusing at first glance and might benefit from the "SOX2" label being position farther down.

Reviewer #4

(Remarks to the Author)

To the reviewer:

Reviewer 1 –

The authors describe a minor modification of the popular CUT&Tag method for chromatin profiling of small samples and single cells, showing that it can be used for transcription factors. The high salt washes used with the standard CUT&Tag protocol can dislodge many TFs, and here a slightly modified buffer appears to suffice to prevent loss. Indeed the data presented do show **improved** recovery relative to CUT&Tag performed using the standard procedure, with **improved** efficiency.

Response 1: We thank the reviewer for acknowledging our work. Indeed, only DynaTag, unlike the original CUT&Tag or its modified versions, clearly shows the successful production of libraries from multiple transcription factors (**15 TFs**). We would like to stress that, although **the protocol modification may appear relatively simple if taken individually**, the overall outcome and achieved **advancement is major**, enabling robust mapping of TFs in small samples both in bulk as well as at single-cell resolution. We present now in **Supplementary Fig. 1** that CUT&Tag and modified protocols **are not applicable** for mapping transcription factors (we tested OCT4, SOX2, NANOG, YAP1 and MYC) (please see **Supplementary Fig. 1**). Overall, we argue that we do not only show “improved” recovery with “improved” efficiency, but our method enables the acquisition of biologically relevant results that were not achievable with previous methods.

However, this brief article is missing the kinds of details that someone interested in trying or switching from the current CUT&Tag protocol will need to know. Moreover, the narrowness of the advance, and the fact that some of the protocol modifications have been published by others, makes it unsuitable for a general interest journal like Nature Communications.

Response 2: We apologize for not clearly articulating the key implications of our approach, and why we are confident that DynaTag offers superior performance compared to other currently available TF mapping methods. We agree that the brief format made it difficult to convey the manuscript’s main message. To address this, we have substantially expanded the manuscript to more effectively illustrate the data, results, and impact of our work.

We respectfully disagree with the labelling of DynaTag as a "**narrow**" advance. DynaTag is the first method to robustly map transcription factors at single-cell resolution and in complex samples. Applied to small cell lung cancer (SCLC), one of the deadliest human cancers, it enabled the mapping of 15 TFs and revealed that mutant P53 exhibits gain-of-function occupancy after chemotherapy. Notably, these insights were derived from nuclei of a patient-derived xenograft model, highlighting both the method’s physiological relevance and its potential to open new directions in p53-focused SCLC research.

The key innovation of the DynaTag protocol—compared to CUT&Tag and its derivatives— lies in its unique nuclei treatment and washing procedure, which employs buffers distinct from those used in previous methods such as CoBATCH, ACT-seq, G-quadruplex CUT&Tag,

Reviewer Comments DynaTag

Nano-CT, or the original CUT&Tag protocol. Therefore, we disagree that "*some of the protocol modifications have been published by others*".

For instance, the core innovation of DynaTag is the use of a physiological intracellular salt buffer, whereas G-quadruplex CUT&Tag employs a high-salt (300 mM potassium) buffer. Critically, this and other high-salt buffers reported in the literature are not suitable for transcription factor mapping, as demonstrated by our experimental evidence (see **Supplementary Fig. 1**).

Much more evidence is needed, including controls that clarify just what is the basis of the improvement over such a widely used method that would explain the technical basis for the proposed improvement.

Response 3: Thank you for this suggestion. We address this by providing additional controls beyond IgG, including potential untargeted tagmentation caused by excess pA-Tn5 or the presence of 1 mM MgCl₂ in the DynaTag buffer. None of these conditions showed evidence of untargeted tagmentation—no sequencing libraries were generated (see **Supplementary Fig. 1**).

We also directly address the key component of the DynaTag buffer that preserves specific TF–DNA interactions. Increasing the KCl concentration from 110 mM to 300 mM in the buffer completely abolished library generation (**Supplementary Fig. 1**). This indicates that physiological intracellular potassium levels are critical for maintaining specific TF binding and enabling successful sequencing.

Major

1) The rationale behind the change in buffer is not explained, nor are the various components tested individually. The change in alkali metal ion from 150 mM NaCl to mostly KCl is not really novel, as it's routinely used by several labs for CUT&Tag when profiling G-quadruplexes, which are stabilized by K⁺, but that isn't mentioned nor is the change justified by the authors. The original CUT&Tag protocol cited and used for comparison (Kaya-Okur 2019) has been used for one of the TFs in this study, Sox2, and the improvement here might have been entirely from lowering the ionic concentration and using Mg⁺⁺ during pA-Tn5 binding to increase tagmentation efficiency. **But there's no clue in the manuscript just what is responsible for the improvement.**

Response 4: We have now expanded our explanation of the rationale behind modifying the nuclei handling buffer. DynaTag enables robust and specific TF–DNA interaction mapping while preventing untargeted tagmentation by employing a physiological intracellular salt composition (1 mM MgCl₂, 10 mM NaCl, 110 mM KCl) throughout the entire nuclei preparation and wash procedure (**Supplementary Fig. 1A–D**). In contrast, protocols such as CoBATCH, ACT-seq, and G-quadruplex CUT&Tag use either high potassium concentrations (300 mM KCl) or 150 mM NaCl with additional components. We show that these conditions fail to preserve specific TF–DNA interactions (**Supplementary Fig. 1E–H**). Critically, even ionic conditions with 150 mM or 121 mM NaCl—matching the effective ionic strength of the DynaTag buffer—resulted in untargeted tagmentation in IgG controls (**Supplementary Fig. 1H–J**). Furthermore, increasing potassium from 110 mM to 300 mM in the DynaTag buffer, as done in G-quadruplex CUT&Tag, abolished library generation for all TFs (**Supplementary Fig. 1G**). Importantly, neither the removal of 1 mM MgCl₂ from the DynaTag buffer nor the fixation status impaired library generation (**Supplementary Fig. 1H–J**), further underscoring that the physiological potassium concentration is the key factor for retaining specific TF–DNA interactions and enabling successful library preparation.

Reviewer Comments DynaTag

Finally, performing the tagmentation reaction in ATAC-seq using the DynaTag buffer in place of the conventional tagmentation buffer did not produce ATAC-seq libraries, indicating an absence of untargeted tagmentation. In contrast, replacing the tagmentation buffer with the CUT&Tag wash buffer containing 150 mM NaCl (instead of the standard 300 mM) led to untargeted tagmentation (**Supplementary Fig. 1K**).

Together, these data demonstrate that the use of a physiological intracellular potassium concentration is both novel and critical for the performance of DynaTag and distinguishes it from previously published protocols.

2) The reduction in salt during tagmentation to improve efficiency is well-known, for example CoBATCH and ACT-seq published in 2019 but not cited used similar ionic concentrations. Also, in PMID: 33191916 for bulk and 34250209 for single cells, low-salt tagmentation greatly improved detection of active histone modifications and RNA Polymerase II. **Although this change in protocol was not applied to TFs, the basic strategy of reducing salt to increase tagmentation is not original.**

Response 5.1: We have now systematically compared the nuclei wash procedure of CoBATCH, ACT-seq and G-quadruplex CUT&Tag with DynaTag (see results in **Supplementary Fig. 1**). Only the sample preparation using the DynaTag protocol produced robust and specific TF sequencing libraries. The physiological potassium concentration in the DynaTag buffer preserves specific TF-DNA interactions, unlike buffers with low or high sodium or high potassium concentrations. **We would like to clarify that both approaches mentioned by the reviewer (PMID: 33191916, 34250209) use high-salt nuclei wash conditions (e.g., 300 mM NaCl), which, like standard CUT&Tag, are unsuitable for efficiently retaining TF-DNA interactions.** Hence, a lower concentration than 300 mM NaCl in the tagmentation buffer is not sufficient to capture dynamic targets, such as TFs.

3) The presence of 1 mM Mg⁺⁺ in the DynaTag buffer is puzzling, because even at room temperature, a 1 hour incubation should result in tagmentation. Perhaps that is what is happening, but the rate of Protein A binding to the antibody is fast enough to avoid open chromatin mapping. Controls are needed to make clear just what is happening.

Response 5.2: To directly address the concern regarding potential untargeted tagmentation due to the presence of 1 mM MgCl₂ in the DynaTag buffer, we performed control experiments using an excess of pA-Tn5. These results showed no evidence of untargeted open chromatin mapping (**Supplementary Fig. 1K**). Furthermore, removing 1 mM MgCl₂ from the DynaTag buffer had no impact on the efficiency of TF library generation, indicating that tagmentation does not occur non-specifically under these conditions (**Supplementary Fig. 1H–J**). In contrast, when we reduced the NaCl concentration during the nuclei wash steps in CUT&Tag from 300 mM to 150 mM or 121 mM, we observed clear signs of untargeted tagmentation (**Supplementary Fig. 1D–E**). These results support that the DynaTag buffer conditions—specifically the physiological K⁺ concentration—prevent non-specific activity of pA-Tn5 despite the presence of Mg²⁺.

4) Tagmentation at 10 mM MgCl₂ needs to be explained mechanistically, as this high concentration, especially in light of point 2 above, may have unintended effects. Again, it is hard to understand why exchanging most of the Na⁺ with K⁺ and why Mg⁺⁺ can give such improved results relative to standard CUT&Tag. Controls are needed.

Reviewer Comments DynaTag

Response 6: All CUT&Tag protocols typically use 10 mM MgCl₂ (PMID: 33191916, 34250209), hence there is no concern about unintended effects. We systematically compared various modified versions of CUT&Tag and found that only DynaTag efficiently retained specific TF-DNA interactions, please see **Supplementary Fig. 1**. Initially, we lowered MgCl₂ to 1 mM to approximate physiological Mg²⁺ levels. However, our new data indicate that the key factor in the efficacy of DynaTag is the physiological K⁺ concentration, rather than Mg²⁺ (see **Supplementary Fig. 1G-J**). Thus, 1 mM nor 10 mM Mg²⁺ introduces unintended effects, and replacing 300 mM Na⁺ with physiological intracellular K⁺ was central to generate specific TF maps by DynaTag.

5) The comparisons to CUT&Tag are not sufficiently documented, with just gel images in a supplement. Snapshot comparisons to ATAC-seq is insufficient, and there is the concern that tagmentation of open chromatin will be a problem for some TF antibodies.

Response 7: We systematically analysed the DynaTag and CUT&Tag libraries with standardized high-sensitivity Bioanalyzer quality control measurements (**Supplementary Fig. 1A** and now **Supplementary Fig. 1**). Bioanalyzer measurements of short-read Illumina sequencing libraries are the gold standard for evaluating library quality before sequencing. Bioanalyzer results showed high library yields and expected size distributions for TF DynaTag and the histone CUT&Tag libraries. In contrast, TF CUT&Tag libraries yielded insufficient material for sequencing, making direct comparisons at the sequencing level impossible (**Supplementary Fig. 1A** and now **Supplementary Fig. 1**). Consequently, our data indicate that DynaTag consistently produces libraries for all TF targets. Very high correlations between matched CUT&Tag and DynaTag data sets confirmed an overall high similarity between these data sets targeting CTCF, as well as the two histone marks H3K27me₃ and H3K4me₃ (**Supplementary Fig. 1L**). FRiP score determinations between these methods further suggested a similar high quality in the identified enriched regions by peak calling (**Supplementary Fig. 1M**), suggesting overall that DynaTag produces similar mapping data in comparison to CUT&Tag.

To address concerns about antibody specificity and potential untargeted tagmentation due to their interaction potential with open chromatin, we used only ChIP-grade antibodies with validated datasets in ENCODE or ChIP-Atlas. We now discuss this point in detail in the revised manuscript (**Fig. 1**). Our approach was based on the rationale that if antibodies bind nonspecifically to open chromatin, all TFs would show similar occupancy across the same accessible regions. Conversely, TF-specific antibodies should yield distinct occupancy profiles with partial overlap only where co-binding occurs. To investigate this, we analyzed occupancy patterns of five TFs (OCT4, SOX2, NANOG, MYC, YAP1) in ESCs. Differential occupancy analysis revealed six distinct categories (OSNY_M, OSNY_M, OSNY_M, OSNY_M, OSNY_M, OSNY_M), confirming that each TF displays specific binding patterns (**Fig. 1E**). This argues against widespread untargeted tagmentation and supports the specificity of the antibodies used.

In addition, DynaTag signal at transcription start sites (TSS) of known target genes (from ChIP-Atlas) further confirmed expected binding patterns for all five TFs (**Supplementary Fig. 2**), validating antibody performance.

Finally, to compare DynaTag data quality with existing methods, we analyzed motif enrichment in MACS2-called peaks from matched DynaTag, ChIP-seq, and CUT&RUN datasets for OCT4, SOX2, and NANOG. The expected known motifs were more enriched in DynaTag than in CUT&RUN and at similar levels to ChIP-seq, further supporting the specificity of the antibodies in the DynaTag approach (**Fig. 1F**).

Reviewer Comments DynaTag

6) Although this protocol is applied to lightly fixed nuclei, the cited paper used native nuclei and later versions used lightly fixed and whole cells and whether the protocol applies to these variations is unclear. This will be important if the new protocol is to be widely adopted.

Response 8: We applied DynaTag to lightly fixed nuclei from cultured cells and frozen cancer tissues. To address the applicability of DynaTag for native nuclei, we have now compared unfixed versus lightly fixed nuclei. Both conditions work equally well, please see **Supplementary Fig. 1H-J**.

7) For such a minor tweak of the CUT&Tag protocol, which is used in hundreds of laboratories and is the basis for multiple kits, calling this DynaTag is not justified. Nor is a new name in the authors' interest, considering that if adopted, the same reagents and kits will suffice, so presenting this as an easily implemented improvement rather than a new method means that investments already made by industry and their customers can be taken advantage of. CoBATCH and ACT-seq are examples of CUT&Tag-like methods that were never adopted, and it would be unfortunate if what looks to be an improvement in a popular method were to be ignored even though the modifications are so simple. Maybe describe it as CUT&Tag with DynaTag buffer instead.

Response 9: We respectfully disagree that our method represents “a minor tweak”. Although the buffer change and nuclei handling might appear simple, it opens-up transcription factor mapping in limited samples or at single-cell resolution—capabilities that neither CoBATCH, ACT-seq nor other modifications of CUT&Tag achieve. As shown in our new data (**Supplementary Fig. 1**), a direct comparison of the DynaTag nuclei wash procedure with other CUT&Tag modifications clearly demonstrates its superior performance. Therefore, we believe that DynaTag should be recognized as a distinct, robust stand-alone method rather than a minor refinement of traditional CUT&Tag protocols, which do not robustly work for TFs.

Our view is supported by pioneer opinions on the current scope of CUT&Tag and modified versions: **PMID: 38129675**  “*Nano-CT has not been tested extensively for non-histone proteins such as TFs, histone modifiers, remodelers or others. Similar to MultiCUT&Tag17, scNTT-seq19 and bulk CUT&Tag9,36, nuclei wash and tagmentation are performed under stringent 300 mM salt conditions to avoid nonspecific tagmentation of open chromatin regions9. Because of the stringent conditions, profiling of weakly bound or less-abundant DNA-associated proteins and TFs might be challenging with the current protocol. Although we have successfully profiled OLIG2 and RAD21 transcription factors by scCUT&Tag11, further work will be required to enable robust profiling of TFs by nano-CT.*”

PMID: 32913232  “*...the requirement for more stringent washes to avoid binding to and tagmentation of accessible DNA also reduces occupancy of transcription factors (TFs)...*”

Reviewer 1 – minor issues:

1) There have been many modifications to the original CUT&Tag protocol from 2019, which is referred to as the protocol used for comparison, making just a reference to the original paper insufficient.

Response 10: We have now referenced other modifications of CUT&Tag and systematically compared their wash procedures with DynaTag, emphasizing its unique ability to map transcription factors (**Supplementary Fig. 1**).

2) The GEO entry number is unrecognized by GEO.

Reviewer Comments DynaTag

Response 11: We have confirmed that the entry number is correctly recognized, in order to access the data the reviewer needs to use the following token: “atipcocozzujvex“

3) For such a brief methods paper, a step-by-step protocol would be helpful, otherwise it is unlikely to be adopted.

Response 12: We agree with the reviewer and will publish a step-by-step protocol in protocol exchange or [protocols.io](https://www.protocols.io) together with the accepted manuscript so that the method can be widely applied.

Reviewer 2 – Major

The manuscript “DynaTag for efficient profiling of transcription factors in small samples and single cells” presents an approach for the improved profiling of protein-DNA interactions, particularly transcription factors (TFs). This is interesting, however this manuscript encompasses 4 major topics, each of which requires substantial additional analysis and documentation to affirm their claims.

First, the authors describe their desire to profile transcription factors by tagmentation, and describe this as “elusive”. They show data on a panel of transcription factor antibodies that do not work in their hands by CUT&Tag, but do work with a modified protocol. The manuscript does not address the rationales for their modifications, or how these have improved profiling. This should be discussed in the text.

Response 13: We have clarified in the main text that high salt wash conditions can disrupt transcription factor (TF)-DNA interactions. To explore this further, we compared our physiological DynaTag wash protocol with previously published CUT&Tag adaptations (ACT-seq, CoBATCH, G-quadruplex CUT&Tag). As shown in **Supplementary Fig. 1** (also now in **Supplementary Fig. 1**), increasing the KCl concentration above 110 mM abolished TF DynaTag library yields, indicating that maintaining a near-physiological KCl level is critical for preserving TF–DNA interactions. Based on these results, we conclude that the physiological potassium concentration used in DynaTag is the key factor enabling robust TF profiling from small samples and at single-cell resolution.

It is critical that these results be distinguished from general accessibility of genomic sites, which is not adequately demonstrated here. Looking at the details of their protocol modifications, the differences seem to be a different monovalent salt in many buffers, a lower concentration of monovalent salts, and the presence of magnesium in all buffers. **It has been previously documented that conditions like these allow untargeted tagmentation (PMID: 32913232).** Figure 1D attempts to address this with selected examples of factor profiling, but a more comprehensive analysis is needed to determine if and how much of profiling signal is explained by accessibility.

Response 14: We addressed these important concerns also in **Response 5.2** and **7** to reviewer 1, and we are happy to repeat here our response: To address potential untargeted tagmentation, we have included multiple controls. First, our IgG-only controls do not generate any sequencing libraries, indicating a lack of non-specific tagmentation. Second, we have tested an excess of pA-Tn5 in the DynaTag buffer (**Supplementary Fig. 1K**); again, no library was produced, underscoring that DynaTag does not enable untargeted tagmentation. In contrast, we show that previously reported CUT&Tag variants (**PMID: 32913232**) exhibit untargeted

Reviewer Comments DynaTag

tagmentation and critically reduced transcription factor library yields (**Supplementary Fig. 1B-E**).

Additionally, we have extensively analyzed the occupancy of five transcription factors in ESCs (**Fig. 1**). We generated cross-comparisons to matched publicly available CUT&RUN and ChIP-seq datasets. These results show that DynaTag yields TF binding landscapes that accurately reflect the binding pattern for each TF in ESCs, with no evidence of untargeted tagmentation (**Fig. 1** and **Supplementary Fig. 1**).

Additionally, it is not clear that the presented ATAC data will be adequate; the shown examples have very small scales, and the other subpanels have widely different scales.

Response 15: In the original **Fig. 1D**, ATAC-seq data were presented for six different open chromatin regions, each normalized to the maximum intensity of the peak. By contrast, the other tracks showed DynaTag data adjusted to the same scale. Because these two methods have different technical biases -open chromatin (ATAC-seq) versus TF binding (DynaTag)- their signal intensities cannot be directly compared. We include now a new broader IGV view for one particular genomic region (**Fig. 1D**) for which several DynaTag and ATAC-seq peaks are shown. Importantly, we have now systemically identified genomic regions that show unique occupancies for at least one of the TFs using differential occupancy analysis (**Fig. 1E**), see methods for detail. This systematic assessment revealed at least 6 genomic categories in which unique differential occupied regions are prevalent in ESCs among the five TFs, NANOG, SOX2, OCT4, YAP1 and MYC.

The motif enrichment analysis presented in Figure 1E shows very poor enrichment of expected motifs (10X relative to random), suggesting that many of the called peaks actually lack high-quality motifs and are not expected binding sites. A more thorough analysis of signal in peaks compared to published profiling should also be included.

Response 16: We have addressed the concern of the reviewer by conducting a comprehensive comparison of transcription factor (TF) mapping data obtained via DynaTag, CUT&RUN, and ChIP-seq for three TFs (SOX2, OCT4, and NANOG) in murine embryonic stem cells. To eliminate biases from differing data processing pipelines, we reprocessed all datasets from FASTQ files using identical software versions and peak-calling parameters, ensuring a consistent and unbiased comparison across methods.

Next, we evaluated TF occupancy signals at transcription start sites (TSS) of consensus target genes for each TF, as annotated in the ChIP-ATLAS database. We present this data at 2 resolutions (left side: TSS +/- 0.5 kb, right side: TSS +/- 2.5 kb). While all three approaches revealed TF occupancy at the expected TSS, DynaTag consistently exhibited higher signal-to-background ratios and superior resolution compared to CUT&RUN and ChIP-seq (**Fig. 1F**). To directly address the concern regarding motif enrichment in DynaTag-identified peaks, we used the latest version of HOMER2, a state-of-the-art tool for motif analysis, to quantify the enrichment of high-quality TF motifs. We identified peaks in all datasets using MACS2 under uniform parameters and then compared motif enrichment among DynaTag, CUT&RUN, and ChIP-seq data (**Fig. 1F**). Notably, DynaTag showed higher enrichment of the expected high-quality TF motifs, which ranked among the top 10 most enriched for each TF. This enrichment was significantly greater than in CUT&RUN and similar to ChIP-seq, underscoring the quality of DynaTag.

In summary, our findings demonstrate that DynaTag not only provides high-resolution TF occupancy mapping with the strongest signal-to-background ratios, but also identifies peaks

Reviewer Comments DynaTag

that show robust enrichment of the expected TF motifs. These results collectively validate the high-quality performance of DynaTag in profiling transcription factor binding sites.

Supplementary Figure 1D should also be expanded to consider signal. This section should also address the mechanism by which these buffer changes are acting. The authors attribute this to retaining factor-chromatin interactions during sample preparation, but this statement is unsupported.

Response 17: We expanded the manuscript and include now a whole section on this topic, which is described in **Fig. 1** and **Supplementary Fig. 1**. Through systematic testing of various wash conditions, we identified that maintaining a near-physiological potassium concentration (110 mM KCl) in the DynaTag buffer is critical for preserving TF–DNA interactions and achieving successful TF library preparation (**Fig. 1, Supplementary Fig. 1** and **Supplementary Fig. 1**). Higher salt concentrations (>110 mM KCl) disrupted these interactions, leading to reduced library yields for most TFs, except CTCF, which is known to bind DNA with a particularly high affinity. This finding suggests that wash buffers used in CUT&Tag and related protocols weaken TF–DNA interactions, leading to TF dissociation from DNA during in vitro nuclei treatments prior to tagmentation.

To further illustrate these effects, we have expanded **Supplementary Figure 1** to include read coverage profiles and tornado plots for OCT4, SOX2, NANOG, MYC, and YAP1. These data show that the DynaTag buffer maintains robust TF binding signals, supporting our conclusion that near-physiological salt conditions are necessary to preserve TF–DNA interactions during sample preparation.

The second topic covered here is a study of ESCs and differentiated epiblast-like cells. The changes in FRiPs and in UMAP counts is marginal between these two cell types, and it is not clear to me that there is really any difference. A more thorough analysis focusing signal at known peaks that change between these cell types for these factors is needed. I question whether the TF binding profiles should really be that different between these very similar cell types.

Response 18: We appreciate the questions of the reviewer regarding differences between ESCs and EpiLCs. First, we clarify that FRiP (fraction of reads in peaks) assesses mapping quality by indicating the proportion of reads aligning within identified peaks. It is not designed to measure differential TF occupancy between cell types. Thus, the similar FRiP scores between ESC and EpiLC libraries indicate both datasets are of high quality.

To directly assess TF occupancy changes, we conducted a differential occupancy analysis at known peaks (**Fig. 2A–C**). Despite the close similarity between ESCs and EpiLCs, our analysis revealed distinct binding profiles for all five TFs studied. To further support these findings, we retrieved TF peaks from the ChIP-Atlas database for both ESCs and EpiLCs and repeated our coverage analysis. This independent analysis also demonstrated decreased NANOG binding in EpiLCs and both increased and decreased occupancy for other TFs (**Supplementary Fig. 3**). These occupancy trends align with observed changes in TF protein levels in chromatin fractions (**Fig. 2E**) and the elevated activity of MYC target genes identified by RNA-seq (**Fig. 2F**).

Regarding UMAP counts, our initial manuscript primarily focused on snDynaTag data from ESCs. To address whether DynaTag can detect occupancy differences at single-cell resolution, we systematically analyzed NANOG, MYC, and OCT4 occupancy in both ESCs and EpiLCs (**Fig. 3A–E**). Aggregated single-cell reads clearly showed decreased NANOG occupancy in EpiLCs, consistent with bulk DynaTag findings (**Fig. 3C–E**). Additionally,

Reviewer Comments DynaTag

pseudobulk aggregated snDynaTag data correlated strongly with bulk read coverage (**Fig. 3C**). After removing nuclei with low read counts (bottom 2%) and normalizing intra-sample heterogeneity (z-score normalization; see **methods** and **supplementary script**), we found a notably higher median read count per nucleus for NANOG in ESCs compared to EpiLCs, confirming NANOG's reduced activity in EpiLCs at single-cell resolution (**Fig. 3F**). UMAP visualization further revealed at least two distinct clusters per TF, with MYC and OCT4 clusters showing similar read counts across ESCs and EpiLCs. However, NANOG occupancy was notably lower in the cluster predominantly composed of EpiLC nuclei (**Fig. 3G–H**). Importantly, each cluster was enriched predominantly with nuclei from either ESC or EpiLC, underscoring that DynaTag effectively captures biologically relevant TF occupancy differences between these closely related cell types at both bulk and single-cell resolutions.

The third topic is a single-cell study of transcription factors in ESCs. This work is not adequately described or analyzed. It is standard in this field to report the quality of this data, including platforms, cell numbers and quality metrics, and these results should be included in the main. The supplementary data shows some evidence for cluster separation but with very poor distinction, raising questions about what these clusters mean.

Response 19: We have substantially expanded our single-cell DynaTag analyses, now presented in a dedicated section (**Fig. 3**). Specifically, we include detailed quality metrics and methodology, such as reads-per-nucleus distributions, the platform used (iCELL8cx with imaging-based single-nucleus detection), and how we filtered peaks to retain only those that show significant changes in occupancy between ESCs and EpiLCs across individual nuclei. Moreover, we provide an output file from the iCELL8cx platform, verifying which wells contained single nuclei that proceeded to PCR library preparation. These comprehensive analyses underscore the robustness of our approach and clarify the biological significance of the observed cluster separations (**Fig. 3**).

A previous study documented CUT&Tag transcription factor profiling in single cells, and should be referred to here (PMID: 33846645). This section should be substantially expanded.

Response 20: We have cited now the mentioned study. Importantly, the authors of PMID: 33846645 mention in a follow-up protocol publication (PMID: 38129675) that their method is not suitable to robustly map transcription factors in bulk and at single-cell resolution.

“Limitations:

Nano-CT has not been tested extensively for non-histone proteins such as TFs, histone modifiers, remodelers or others. Similar to MultiCUT&Tag17, scNTT-seq19 and bulk CUT&Tag9,36, nuclei wash and tagmentation are performed under stringent 300 mM salt conditions to avoid nonspecific tagmentation of open chromatin regions⁹. Because of the stringent conditions, profiling of weakly bound or less-abundant DNA-associated proteins and TFs might be challenging with the current protocol. Although we have successfully profiled OLIG2 and RAD21 transcription factors by scCUT&Tag11, further work will be required to enable robust profiling of TFs by nano-CT.”

The fourth topic is a study of small cell lung cancer xenograft and recurring tumors. This section lacks important documentation and analysis. It is not stated how many samples were analysed, what their relationship is, how many cells, what is the quality of the data and results.

Reviewer Comments DynaTag

Response 21: We apologize for the confusion. We have now added the details on how many tumors from how many animals were analysed by DynaTag and snRNA-seq, see Methods – “DynaTag” and “Single-nucleus RNA-seq and Gene Set Enrichment Analysis”. We also substantially increased evaluation of cluster qualities to assess the robustness of the cancer-derived clusters (please see **Supplementary Fig. 4**).

Supplementary Figure 4D shows very poor segregation of multiple markers between the presumptive clusters, and an independent method of assigning or verifying cluster identity is needed.

Response 22: We apologize for the misleading color coding in the original **Supplementary Fig. 4D**, now **Supplementary Fig. 4F**, which made segregation appear poor. After adjusting the color scale and increasing the resolution of clustering until a maximal cluster stability was reached, which we evaluated by repetitive iterations (n =20 per chosen resolution) at different resolutions and assessing cluster similarities, calculating mean Jaccard Indexes per cluster, it is now clear that 10 clusters show distinct segregation, including the two murine immune and stromal clusters (**Supplementary Fig. 4C**). We further include now the expression level of the top 10 distinctly expressed genes per cluster to underline cluster identities (**Supplementary Fig. 4D**).

This section includes profiling of P53 which could be particularly of interest, and for this reason it should be assessed whether the antibodies used here for TFs detect the intended protein; in other words, do they give no peaks in cell types or settings without the target protein?

Response 23: We used a p53 antibody (DO-1) that is ChIP-grade and knockout-validated by Abcam (see product information). To confirm its specificity in detecting the gain-of-function p53 mutation R248Q present in our SCLC PDX model (supporting information), we hypothesized that R248Q would exhibit distinct binding behaviour compared to wild-type (WT) p53.

We identified two independent ChIP-seq datasets—one mapping WT p53 in NSCLC via exogenous overexpression (PMID: 33473123), and another mapping endogenous R248Q p53 in NSCLC (GSE253478). Both datasets were reprocessed from raw FASTQ files using the same analysis pipeline as DynaTag to ensure consistency. We then compared the normalized occupancy signals for WT and R248Q p53 across their respective peaks. This revealed largely non-overlapping regions bound by each p53 variant, suggesting that R248Q p53 has a distinct DNA-binding profile consistent with a gain-of-function phenotype in NSCLC (GSE253478).

Importantly, p53 occupancy in our SCLC PDX samples—measured by DynaTag both before and after chemotherapy—displayed strong enrichment in the R248Q-specific peaks but no enrichment in WT p53 peaks. Likewise, WT p53 data showed no signal in the SCLC PDX peaks, which NSCLC R248Q p53 instead occupied. These observations confirm the expected gain-of-function occupancy pattern of R248Q p53 in SCLC, thereby validating both the specificity of the DynaTag method and our choice of p53 antibody.

Reviewer 2 – Minor

The justification behind naming the approach “dynaTag” is unclear. It doesn’t measure the dynamics of chromatin proteins, and previous publications have described the profiling of transcription factors. Perhaps consider a name that represents what about this protocol gives improved results.

Reviewer Comments DynaTag

Response 24: We politely disagree. In our understanding, weak-interactions of transcription factors are considered as dynamic interactions, also called dynamic weak interactions or transient weak interactions (PMID: 34001530, 25368424). We define DynaTag as a method that can map dynamic targets that exhibit dynamic, weak interactions with DNA. These interactions are more sensitive to high-salt conditions, which is the reason why CUT&Tag and previously modified protocols perform poorly on dynamic targets, see also limitations of CUT&Tag (PMID: 32913232 “...*the requirement for more stringent washes to avoid binding to and tagmentation of accessible DNA also reduces occupancy of transcription factors (TFs)...*”)

The motivation behind comparing DynaTag to using ATAC-seqs alone to identify transcription factor binding is unclear. Previously ATAC-seq is used alongside motif analysis for estimated binding, Is this the motivation behind this hypothesis? Clarification would help with the flow of experiments.

Response 25: We apologize for making our intention not clear enough. We explored whether differential occupancy trends measured by DynaTag could be validated independently through ATAC-seq. While ATAC-seq can predict transcription factor occupancy by quantifying motifs in accessible chromatin or assessing changes in local read depth around those motifs (TF footprints), DynaTag directly measures TF binding. **Our side-by-side comparison showed that TF footprints derived from ATAC-seq does not accurately predict the changes in TF occupancy observed by DynaTag in our ESC–EpiLC comparison.** This highlights the importance of directly measuring transcription factor occupancy, rather than relying solely on accessibility-based predictions, especially for high-resolution analyses at both bulk and single-cell levels.

Figure 1i is not labeled, and the legend describes it as “snDT”, presumably scDT?

Response 26: We apologize for the mistakes and provide corrections in the updated manuscript.

Previous publications have presented profiling of transcription factors by CUT&Tag, including the original publication which included profiling of CTCF, NPAT, and Sox2 in human cells, and other papers have presented profiling of Olig2 and of Rad21. Why do the authors think that these previous CUT&Tag experiments have worked while theirs do not?

Response 27: We would like to clarify that our CUT&Tag libraries for CTCF, which is a benchmark high-affinity transcription factor that retains bound to DNA despite high-salt nuclei wash treatments, correlated well with the CTCF DynaTag libraries (**Supplementary Fig. 1**). In our hands, and using high-salt (300 mM NaCl or KCl) nuclei wash treatments, murine SOX2 profiled by CUT&Tag in the murine ESC system were hardly detectable by Bioanalyzer (High-sensitivity DNA kit), hence, insufficient for sequencing. In the original CUT&Tag paper human SOX2 was profiled in a human ESC cell line (H1). Human in contrast to murine SOX2 could be more resistant to high-salt conditions, potentially explaining why human SOX2 in human ESC worked by CUT&Tag. Hence, CUT&Tag works for a very small set of transcription factors, and importantly, it is stated by the pioneers and drivers of the technology, including the Henikoff lab that produced SOX2, CTCF and NPAT maps by CUT&Tag, that CUT&Tag and modified versions *are not suitable* for the study of dynamic targets, such as transcription factors (PMID: 32913232, 38129675). (PMID:

Reviewer Comments DynaTag

32913232 “...the requirement for more stringent washes to avoid binding to and tagmentation of accessible DNA also reduces occupancy of transcription factors (TFs)...”); “Limitations: Nano-CT has not been tested extensively for non-histone proteins such as TFs, histone modifiers, remodelers or others. Similar to MultiCUT&Tag17, scNTT-seq19 and bulk CUT&Tag9,36, nuclei wash and tagmentation are performed under stringent 300 mM salt conditions to avoid nonspecific tagmentation of open chromatin regions9. Because of the stringent conditions, profiling of weakly bound or less-abundant DNA-associated proteins and TFs might be challenging with the current protocol. Although we have successfully profiled OLIG2 and RAD21 transcription factors by scCUT&Tag11, further work will be required to enable robust profiling of TFs by nano-CT.”.

Reviewer 3 – Major

1. Genome tracks depicting data generated via DynaTag **should** be shown in the main figures. The method is exclusively described by aggregative statistics over classes of peaks, with the exception of several small areas in figure 1D; example tracks of data generated by this method, over entire gene loci etc. should be shown to allow for comparison of the data to existing methods.

Response 28: We appreciate the suggestion to include genome track visualizations of DynaTag data in the main figures. We have now incorporated broader IGV genome browser views at representative loci, along with tornado plots systematically comparing TF occupancy across all significant categories identified by differential occupancy analysis (**Fig. 1D–E**). Additionally, broader genome browser views comparing bulk DynaTag, aggregated single-nuclei, and single-nuclei DynaTag data for MYC, OCT4, and NANOG have been included in the main **Fig. 3A**. Finally, we have added a dedicated section explicitly comparing the performance of DynaTag to established methods, including CUT&RUN and ChIP-seq, to clearly demonstrate the utility and robustness of our approach (**Fig. 1F**).

2. Related to point (1), the data for the scDynaTag as tracks for single cells should be shown. The data in figure 1I is not interpretable; the manuscript mentions the UMAP reveals "heterogeneity" within the population, but there are only two clusters in each UMAP and not enough data is presented to see that this heterogeneity represents biology and not technical artifacts. Can scDynaTag distinguish ESC from EpiLC, for instance?

Response 29: As noted in **Response 28**, we have expanded **Fig. 3** to include broader genome browser views of our single-nucleus DynaTag (snDynaTag) data. We apologize for the initially limited presentation. In this revised manuscript, we generated matched ESC and EpiLC snDynaTag datasets for MYC, OCT4, and NANOG. Our results show clear differential occupancy patterns between ESCs and EpiLCs at the single-cell level, both in UMAP space and at pseudobulk resolution. In particular, the UMAP plot for each transcription factor displays two main clusters, each predominantly composed of either ESC or EpiLC nuclei (**Fig. 3H**). These observations confirm that snDynaTag effectively discriminated between even closely related cell types, corroborating our bulk findings (see also **Response 18**).

3. Related to points (1) and (2), the data in figure 2C should be presented as tornado plots rather than aggregate traces over classes of loci, to allow for visualization of individual peaks.

Reviewer Comments DynaTag

Response 30: In our original manuscript, we used averaged aggregated profiles over the relevant gene sets in **Figure 2C**. Because each gene set contains only 10–100 genes, tornado plots can appear noisy due to the limited number of data points. Nonetheless, we have already assessed the significance of coverage changes across individual promoters (from -1 kb to +200 bp) before and after chemotherapy (**Fig. 2D**). This analysis evaluates read coverage changes on a per-promoter basis, thereby supporting the aggregated trends in **Figure 2C**. To address the reviewer's request, we now provide tornado plots in **Response Fig. 1**, offering an additional visualization trend across individual peaks. We prefer to show the aggregated plots in combination with the significance analysis of the changes in the bubble plot (**Fig. 2D**).

Reviewer Comments DynaTag

Response Fig. 1: Heatmaps of coverage profiles shown in figure 2C as well as ratio of coverage (Chemo vs Control, log₂ CPM).

4. More validation should be performed comparing DynaTag data with the ATAC-seq data; previous reports (PMID 32913232) have demonstrated that lower salt washes can lead to off-target pA-Tn5 insertions at accessible sites. Could the authors show something like tornado plots for their CUT&Tag vs. DynaTag vs. ATAC-seq data?

Response 31: To address whether the low salt condition in the DynaTag buffer leads to untargeted Tn5 tagmentation, please see our new control experiments in **Supplementary Fig. 1**, which collectively showed that DynaTag buffer does not lead to untargeted tagmentation. For example, IgG controls did not produce libraries and an artificially enforced high access of pA-Tn5 in the DynaTag buffer did not result in libraries.

5. In figure 1E, what are the relative rankings of the discovered motifs? Do these motifs represent the most significant or well-represented motifs from each analysis?

Response 32: As described in **Response 16** (see also new **Fig. 1**), we performed a comprehensive, parallel motif enrichment comparison for matched DynaTag, CUT&RUN, and ChIP-seq data. For each TF profiled with DynaTag, the expected motifs consistently appeared among the top 10 enriched motifs. Moreover, DynaTag data showed higher motif enrichments, superior resolution and signal-to-background ratios at the TSS of known target genes compared with the matched CUT&RUN datasets.

Reviewer 3 – Minor

1. More information regarding comparisons of DynaTag to CUT&RUN should be added to the manuscript. CUT&Tag is not efficient at capturing TF interactions, but CUT&RUN works well at profiling these interactions and is the more relevant comparator to DynaTag. The only point in the manuscript that mentions CUT&RUN is the last sentence which briefly mentions it needs more cells.

Response 33: We thank and appreciate the acknowledgment of the reviewer that CUT&Tag is not suitable to profile TFs. We now introduced a systematic comparison between DynaTag, CUT&RUN and ChIP-seq for data sets that matched our system (murine ESCs) and TFs. The analysis is presented in **Response 16** and in **Fig. 1**.

2. Figure 1A is not informative; elements describing the novel differences of DynaTag that allow for TF profiling should be added.

Response 34: We thank this reviewer for pointing out that the scheme does not show the essence why the physiological nuclei wash condition retains transcription factor interactions. We have now modified the scheme to explicitly show the key ingredients that mimic the physiological intracellular salt condition. We have also included a schematic part that shows that the DynaTag buffer is not leading to untargeted tagmentation, which was our key concern and was raised by all reviewers.

3. Throughout the manuscript, mouse proteins should have only the first letter capitalized, rather than all letters capitalized (which is the convention for human proteins).

Reviewer Comments DynaTag

Response 35: According to the nomenclature of Nature guidelines (<https://www.nature.com/nrm/for-authors/preparing-your-submission>), mouse protein name letters are all capitalized.

4. Figure 1I is missing a letter label in the figure.

Response 36: We apologize for this mistake, we changed this. The new UMAP can now be found in Figure 3F-G.

5. The color scales in figures 1I and S4 are too similar to clearly see differences.

Response 37: We adopted the colour scheme. You can find the new data now in figure 3 and supplementary figure S5.

To the reviewer 2:

Reviewer 2 –

Comment 1: “For this reason, we generated the DynaTag physiological salt buffer containing 110 mM KCl, 10 mM NaCl and 1 mM MgCl₂. This cation buffer composition is based on electrophysiological salt concentration measurements *in situ*²⁰, and thereby ensures the retainment of specific TFDNA interactions during sample preparation (Fig. 1a).” This buffer seems to be the key innovation, but I’d like the authors to say more about this. It is not clear to me how this buffer would affect TF retention in **formaldehyde-fixed nuclei**. There is no data here that TFs are being lost from samples in other buffers, only that profiling works in the KCl buffer. The authors should consider literature describing that Na⁺ and K⁺ ions have **different effects on chromatin compaction**, and that this **may affect antibody accessibility or tagmentation**. This could be included in the Discussion.

Response 1: Thank you for the feedback. We would like to clarify that the CUT&Tag and DynaTag protocols use mild fixation (0.1% formaldehyde for 1–2 minutes), unlike ChIP-seq, which typically uses 1% for up to 15 minutes. Such mild fixation does not efficiently crosslink TFs to DNA (PMID: 22955991), allowing for TF dissociation. We include this step to reduce nuclei aggregation and sample loss.

Regarding the DynaTag buffer, our previously revised manuscript (NCOMMS-24-56873A-Z) includes Supplementary Fig. 1 demonstrating that the DynaTag buffer enables specific Histone and TF library preparation while preventing non-specific Tn5 activity (untargeted tagmentation). Unlike sodium (150 mM Na⁺), which causes non-specific tagmentation (Supplementary Fig. 1K), physiological potassium (110 mM K⁺) in the DynaTag buffer does not lead to non-specific Tn5 DNA interactions. This is consistent with other literature showing that Na⁺ destabilizes DNA Histone interactions in comparison to K⁺, suggesting that K⁺ is more physiologically relevant (PMID: 25688036, 35177143).

In summary, sodium can disrupt protein–DNA interactions, while potassium preserves them, supporting our model. However, we argue that a systematic study on the specific effects of Na⁺ vs. K⁺ on TF–DNA interactions lies beyond the scope of this work and should therefore not be discussed in this manuscript.

Comment 2: Figure 1: How many sites are represented in each box in Figure 1E. Is each box different numbers of sites? This is crucial to document how many sites overlap between the profiled factors.

Response 2: Thank you for raising this. As written in the figure legend 1E:”Tornado plots of the DynaTag occupancies (OCT4, SOX2, NANOG, MYC, YAP1) in the six sets of regions (OSNY_M, OSNY_M, OSNY_M, OSNY_M, OSNY_M, OSNY_M) revealed by DOA in ESCs. All boxes in columns use the same set of regions represented either by OSNY_M, OSNY_M, OSNY_M, OSNY_M, OSNY_M or OSNY_M. We have now added in the revised figure legend the total number of regions

within each set of regions: ...” *e*, Tornado plots of the DynaTag occupancies (OCT4, SOX2, NANOG, MYC, YAP1) in the six sets of regions OSNY_M(23,420), OSN_{YM}(28,681), OSN_{YM}(2,613), OSN_{YM}(1,857), OSN_{YM}(1,504) and OSN_{YM}(5,207) revealed by DOA in ESCs. Occupancy is shown as counts per million for 2.5 kb up- and downstream of the peak centres...”

Comment 3: Is it expected that the Kdm3B site shown in Figure 1D would bind $\frac{1}{2}$ factors, including YAP1?

Response 3: To address this question, we considered peaks from ChIP-Atlas of reproducible ChIP-seq data of NANOG, SOX2 and OCT4 since YAP1 and MYC have not been extensively studied by ChIP-seq in murine stem cells. We found that all factors (NANOG, OCT4, SOX2) reproducibly target this Kdm3B site shown in Figure 1D, validating our observation by DynaTag with external ChIP-seq data.

Response Fig. 1: IGV snapshot of DynaTag ESC normalised coverage (OCT4_DynaTag, SOX2_DynaTag, NANOG_DynaTag) and peaks derived from all available ChIP-seq data sets of OCT4 (OCT4_stem.cells_ChIP.Atlas), SOX2 (SOX2_stem.cells_ChIP.Atlas) and NANOG (NANOG_stem.cells_ChIP.Atlas).

Comment 4: What is the scale on the motif enrichment bar plots: the label says “-log₁₀P” but then is also labeled as exponents. The plots for all three methods should be on the same axis, so that they can be compared. For Oct4 in Figure 1F, it appears that only ChIP-seq recovers the expected motif with significance; the other methods do not meet standard cut-offs.

Response 4: The scale is -log₁₀P as illustrated and labelled as exponents because the significance values for the known motifs can be quite high. For example, instead of labelling OCT4 DynaTag enrichments -log₁₀P = 4080, we wrote 4.08e+03. We don’t think a similar scale for all plots or one combined plot would help the visual comparison, because the known motifs for OCT4 are notably more enriched in the ChIP-seq data than in the CUT&RUN and DynaTag data. As consequence the bars of the CUT&RUN and DynaTag data would not be visible, including the other ChIP-seq, CUT&RUN and DynaTag data. Importantly, we argue that the absolute enrichment value is not the important quality characteristic but the observation that the expected known motifs (red bars) are among the top 10 enriched motifs for each method mapping the respective factor. We would like to emphasise that all three methods (ChIP-seq, CUT&RUN, DynaTag) do recover the expected motif with high significance and meet the

standard cut-offs. OCT4, SOX2, and NANOG are known to co-bind genes and interact on the protein level (<https://doi.org/10.1016/j.cell.2009.01.001>; <https://doi.org/10.1016/j.cell.2005.08.020> ; <https://doi.org/10.1038/srep13533>).

Therefore, we anticipated a strong enrichment of their shared motif, OCT4-SOX2-TCF-NANOG, due to the high activity of these pluripotency TFs in embryonic stem cells. Of note, all three mapped TFs (OCT4, SOX, and NANOG) show a significant enrichment for the cooperative OCT4-SOX2-TCF-NANOG motif regardless of technique. This further highlights the ability of DynaTag to uncover reported binding mechanisms of these pluripotency TFs.

Comment 5: Figure 4E - how many sites in each box? Is there no data for DynaTag profiling in wt?

Response 5: As in Fig. 1E, Fig. 4E shows Tornado plots and boxes in columns contain the same peak set (3 peak sets). To make this clear, we added the name and the respective number of peaks in each peak set to the legend of Fig. 4E. There is no data for DynaTag profiling in wt P53. We observed R248Q P53 binding behaviour in the DynaTag data, which we validated with published R248Q P53 ChIP-seq data in Fig. 4E.

“...e, Tornado plot matrix of differential P53 occupancy for wild type (WT) and R248Q P53 mutant. Rows indicate the ChIP-seq or DynaTag signals, cancer type (SCLC PDX or NSCLC), and sample type (normalised read coverage (cpm)), which were plotted in peaks (WT_NSCLC (3,034), R248Q_NSCLC (3,875)) derived from the different ChIP-seq and DynaTag data sets (columns).”

Comment 6: Figure 3E - $\log_{10} p$ is $\ln(1+x)$?

Response 6: This is correct. We added this to the legend to make it clear.

Comment 7: DynaTag: Low-affinity binding is not the same as on-off dynamics, and the title misrepresents the method: “DynaTag profiles dynamics of...”; it does not measure dynamics, in the sense that live imaging or single-molecule footprinting do.

Response 7: We agree with the reviewer. We changed the title to “DynaTag for efficient mapping of transcription factors in low-input samples and at single-cell resolution”.